# Current and future goals are represented in opposite patterns in object-selective cortex

Anouk Mariette van Loon[1,2†]*, Katya Olmos-Solis[1†], Johannes Jacobus Fahrenfort[1,3‡], Christian NL Olivers[1,2‡]*

[1]Department of Experimental and Applied Psychology, Vrije Universiteit Amsterdam, Amsterdam, The Netherlands; [2]Institute of Brain and Behavior Amsterdam, Vrije Universiteit Amsterdam, Amsterdam, The Netherlands; [3]Department of Brain and Cognition, University of Amsterdam, Amsterdam, The Netherlands

**Abstract** Adaptive behavior requires the separation of current from future goals in working memory. We used fMRI of object-selective cortex to determine the representational (dis)similarities of memory representations serving current and prospective perceptual tasks. Participants remembered an object drawn from three possible categories as the target for one of two consecutive visual search tasks. A cue indicated whether the target object should be looked for first (currently relevant), second (prospectively relevant), or if it could be forgotten (irrelevant). Prior to the first search, representations of current, prospective and irrelevant objects were similar, with strongest decoding for current representations compared to prospective (Experiment 1) and irrelevant (Experiment 2). Remarkably, during the first search, prospective representations could also be decoded, but revealed anti-correlated voxel patterns compared to currently relevant representations of the same category. We propose that the brain separates current from prospective memories within the same neuronal ensembles through opposite representational patterns.
DOI: https://doi.org/10.7554/eLife.38677.001

*For correspondence:
anouk.vanloon@gmail.com (AML);
c.n.l.olivers@vu.nl (CNLO)

[†]These authors also contributed equally to this work
[‡]These authors also contributed equally to this work

**Competing interests:** The authors declare that no competing interests exist.

## Introduction

Adaptive human behavior requires the representation of both imminent and future goals in response to changing task requirements. Little is known about how the brain distinguishes between information that is currently relevant and information that is only prospectively relevant.

While working memory is thought to be pivotal to the active maintenance of representations for current task goals, representations serving prospective tasks should be shielded from affecting currently relevant input and output, and vice versa. Studies using reaction time and eye movement measures have indeed shown that currently and prospectively relevant representations differentially bias processing of perceptual input (e.g., *Carlisle and Woodman, 2011*; *Downing and Dodds, 2003*; *Houtkamp and Roelfsema, 2006*; *Mallett and Lewis-Peacock, 2018*; *Olivers and Eimer, 2011*; *van Loon et al., 2017*). Furthermore, studies using multi-variate pattern analyses (MVPA) of functional magnetic resonance imaging (fMRI) and electroencephalography (EEG) data have shown that while representations required for an upcoming memory test can be readily decoded, the evidence for items required for a prospective task temporarily drops to baseline levels until they become relevant again (*LaRocque et al., 2013*; *LaRocque et al., 2017*; *Lewis-Peacock and Postle, 2012*). These and other findings have led to the hypothesis that items in working memory may adopt different states or representational formats (*Barak and Tsodyks, 2014*; *D'Esposito and Postle,*

2015; *Larocque et al., 2014*; *Olivers et al., 2011*; *Stokes, 2015*). While currently relevant items are represented through patterns of firing across populations of neurons, prospectively relevant items may be stored in what has been referred to as an 'activity-silent', or 'hidden' state. One way in which such a state can be achieved is through short-term potentiation of synaptic connectivity in the neuronal population, as induced by the initial firing activity during encoding and active storage within that same population (*Erickson et al., 2010*; *Mongillo et al., 2008*; *Sugase-Miyamoto et al., 2008*). Another way is through changes in the membrane potentials of the previously firing neurons (e.g., *Stokes, 2015*). We will collectively refer to these options as changes in the responsivity (versus the activity) of a neuronal ensemble.

Such latent changes in responsivity are by definition difficult to test through activity-based measures. One prediction is that prospective memories re-emerge in activity-based dependent measures when *unrelated* activity is sent through the network and interacts with the pattern of changed responsivity that reflects the activity-silent memory. This is indeed what *Rose et al. (2016)* recently reported. They found that prospective memory representations which could initially no longer be decoded during a working memory delay period could successfully be reconstructed after applying a brief burst of transcranial magnetic stimulation (*Rose et al., 2016*). Likewise, Wolff and colleagues recently reported enhanced decoding of a memorized oriented grating shortly after observers were presented with a visual pattern that was neutral with respect to the memorized orientation (*Wolff et al., 2015*; *Wolff et al., 2017*). However, although these studies show that there is information present on prospectively stored memories, it is as yet unclear what the representational format of such prospective memories is, and how they relate to currently relevant memories.

A priori there appear to be a number of hypotheses. First, the standard synaptic potentiation mechanism predicts that the altered pattern of responsivity directly follows the pattern of activity during encoding of the item, thus predicting a high degree of similarity between the active and the silent representation when revived. A second possibility is that it is unnecessary to assume activity-silent representations at all, as has recently been argued by *Schneegans and Bays (2017)*. Instead, they argued for a single maintenance mechanism in which differently prioritized items in memory are stored through similar patterns of firing activity, with the only difference being the degree of activation. Their model simulations provide a proof of concept that the revival of a memory can be explained by selectively boosting the still present, but lowered activity, rather than by the reconstruction from hidden states of responsivity. Also under this scenario the same pattern of activation should emerge for current and prospective memories, except for a difference in strength. The third possibility is that prospectively relevant items are stored in an altogether different pattern compared to actively maintained items – that is, they may be transformed within the same population, or stored in different populations, whether through changed activity or responsivity. This was recently proposed by *Christophel et al. (2018)*, who found currently relevant items to be represented more strongly in posterior brain areas (notably visual cortex), while prospectively relevant items were represented more strongly in frontal regions (notably the Frontal Eye Fields). Under this scenario the representational overlap between current and prospective items within the brain regions involved is expected to be minimal. Although crucial for current theories of working memory, so far, studies have not directly compared the representational pattern of current and prospective memories.

In two experiments, we aimed to further understand how working memory distinguishes between information relevant for either imminent or future goals. We asked observers to perform two consecutive visual searches for particular target objects drawn from different object categories (see *Figure 1A and B* for Experiments 1 and 2 respectively). Prior to search, these objects would be maintained in working memory as target templates. Importantly, a cue indicated whether the target template of interest would be relevant for the *first* search (turning it into a *current* template), for the second search (turning it into a prospective template) or would not be relevant for either search task (irrelevant condition, only in Experiment 2). Using MVPA of fMRI activity in object-selective visual cortex, we directly compared the neural representations of these templates when needed for the current search task, to when needed for the prospective search task. Experiment 1 served to establish the relationship between currently and prospectively relevant representations, while Experiment 2 extended the comparison to representations that could be dropped from memory entirely, as they became irrelevant for the subsequent tasks.

These experiments reveal a dissociation between currently relevant, prospectively relevant and irrelevant templates based on category selective patterns in object-selective cortex. We find that

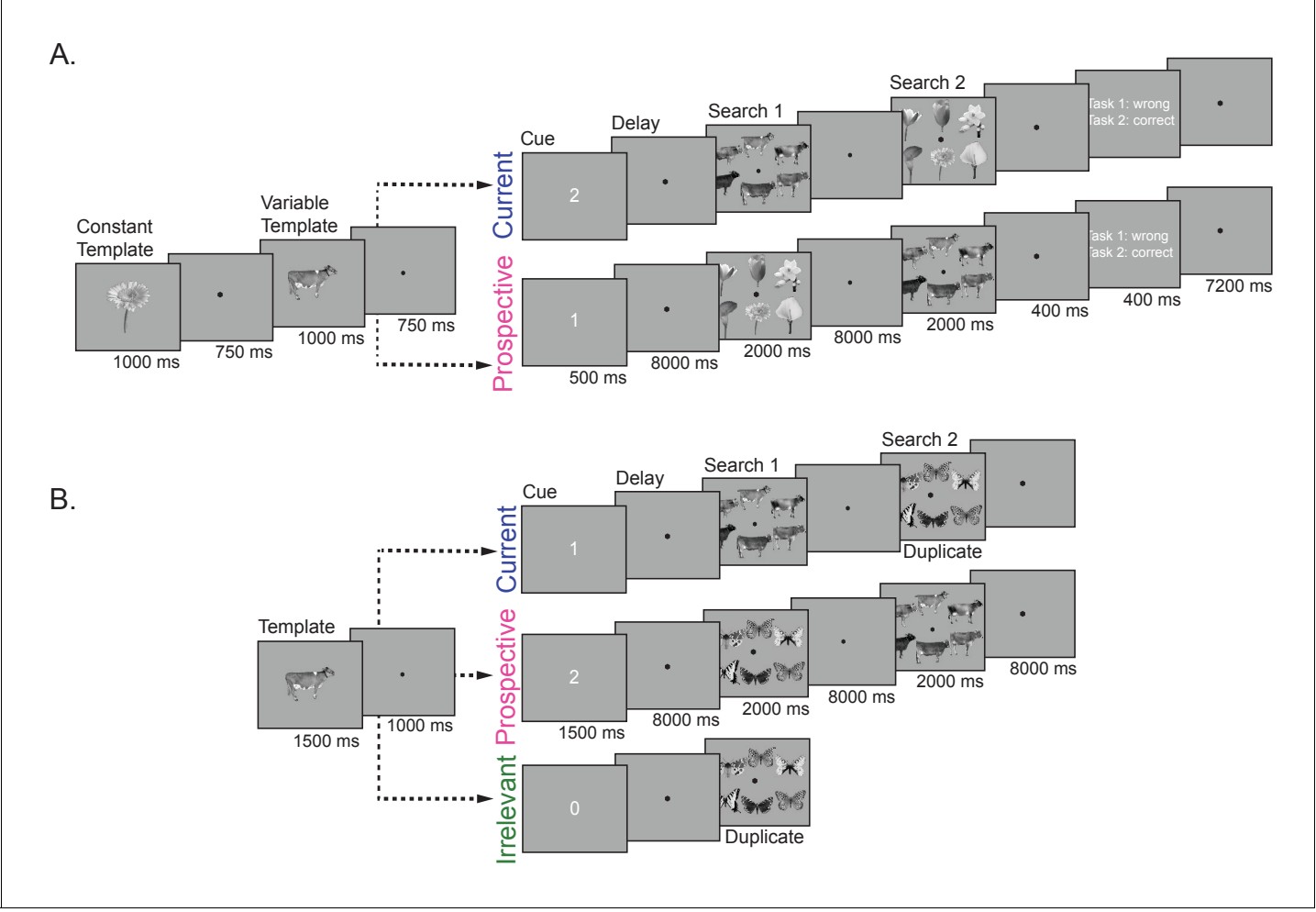

**Figure 1.** Trial design. (A) Experiment 1. On each trial, participants performed two consecutive visual search tasks. The target objects for both search tasks were presented at the start of the trial. One of the objects could either be a cow, dresser or skate (variable template search; four exemplars per category), and was used for the decoding analyses. The other target was always the same flower (constant template search). The order of presentation (constant or variable template) was manipulated between trials, to create the two main conditions – one in which the variable template was currently relevant, the other in which it was prospectively relevant. To this end, a retro-cue ('1' or '2') indicated which of the two previously memorized objects was the target in Search 1. The cue was followed by a delay, then the first search display, followed by a second delay and finally the second search display. Thus, in the Current condition, observers first searched for the variable template (cow, dresser, or skate), and then for the constant template (flower), while this order reversed in the Prospective condition. For each search display, participants indicated whether the target object was present or absent using a button press. At the end of each trial and run participants received feedback about their performance. (B) Experiment 2. Here participants were presented with only one object (cow, dresser or skate) as the possible target template for one of two consecutive visual search tasks. Then a retro-cue appeared, when the cue was '1' the memorized object was a current template, for Search 1; cue '2' indicated that the object was a prospective template, for Search 2; finally, when the cue was '0' the memorized item was not a target in either search and thus it was irrelevant in the trial. The remaining search task in Experiment 2 (either Search 2 in the Current condition, or Search 1 in the Prospective and Irrelevant conditions) was a so-called duplicate search task. In this task, butterflies, motorcycles or trees were presented and participants indicated whether or not any one of the exemplars was shown twice in the display. Thus, here no search template could or needed to be prepared. In the irrelevant condition participants only performed the duplicate search task.

DOI: https://doi.org/10.7554/eLife.38677.002

while observers are searching displays for the current target, the prospective search template can nevertheless be temporarily decoded, extending the demonstration that prospective memories can be reconstructed by sending unrelated activity through the network (*Rose et al., 2016*; *Wolff et al., 2017*). Most importantly, we find that during the first search, the pattern of activity corresponding to the prospective template is the inverse of the same template when it is currently relevant. Thus, patterns reflecting prospective and current memory templates are systematically dissimilar, even

when they belong to the same category. Experiment 2 further demonstrates that this inverse representational code is specific to the maintenance of information for future goals, as irrelevant representations did not show such an inversion. These results suggest that prospective templates are protected from interfering tasks by maintaining them in an opposite representational space.

## Results

### Experiment 1

To examine the relationship between currently and prospectively relevant representations, on each trial observers (N = 24) performed two consecutive visual search tasks (Search 1 and Search 2). The two to-be-sought-for target templates were presented at the start of each trial, after which a cue indicated which of the two targets would have to be looked for first – thus making it currently relevant, while the other target became prospectively relevant. To limit the working memory load, and to maximize the chances of decoding current and prospective targets and their differences (see Materials and Methods), we only varied the target from trial to trial for one of the two searches (thus referred to as the 'variable template search'). These targets served as the basis of the multivariate pattern classification analyses, and could thus either be Current or Prospective in nature. The other search was always for the same flower (referred to as the 'constant template search'). The flower search served as an additional task to assign current or prospective status to the variable template, but the flower itself played no role in the classification analyses. For each search, participants indicated whether the target object was present or absent among six exemplars of the same category.

### Behavioral results

*Table 1* shows the mean RTs and mean accuracy for both types of search task (variable template search and constant template search) when they came either first or second in the trial. These measures were each entered in a two-way repeated measures ANOVA (N = 24) with factors search order (Search 1 and Search 2) and type of search task (constant versus variable template). As expected, the constant template search was overall faster (RT: $F_{(1,23)}$ = 928.18, p < 0.001, $\eta_p^2$ = 0.97) and more accurate (percentage correct: $F_{(1,23)}$ = 183.06, p < 0.001, $\eta_p^2$ = 0.88) than the variable template search. Furthermore, the first search was more accurate than the second ($F_{(1,23)}$ = 10.87, p = 0.003, $\eta_p^2$ = 0.32). There was also an interaction for both accuracy and speed ($F_{(1,23)}$ = 9.22, p = 0.006 $\eta_p^2$ = 0.28 and $F_{(1,23)}$ = 6.32, p = 0.02, $\eta_p^2$ = 0.21 respectively): the variable template search had the lowest accuracy when performed second, while the constant template search was fastest when second. Overall these results indicate that, as intended, working memory was indeed involved more in the variable template than in the constant template task. Our subsequent fMRI decoding analyses are based on the variable template, which could either be currently or prospectively relevant.

**Table 1.** Percentage correct and Reaction Time (RT) for Current and Prospective conditions in Search 1 and Search 2 (N = 24) as a function of search order.

| | Current | | Prospective | |
|---|---|---|---|---|
| | Search 1 | Search 2 | Search 1 | Search 2 |
| Template | Variable | Constant | Constant | Variable |
| P. Correct (%) | 82.2 (7.1) | 98.1 (2.2) | 98.0 (2.3) | 76.0 (9.9) |
| RT (ms) | 1387(20) | 772 (21) | 794 (22) | 1411 (22) |

DOI: https://doi.org/10.7554/eLife.38677.003
The following source data is available for Table 1:
Source data 1. Behavioral data for each participant of Experiment 1.
DOI: https://doi.org/10.7554/eLife.38677.004

## fMRI results: Target template decoding as a function of current and prospective relevance

Our analyses targeted posterior fusiform cortex (pFs) which is known to be involved in representing object categories, and which we independently mapped for each participant (following *Harel et al., 2014*; *Kravitz et al., 2013*; *Lee et al., 2013*; *Malach et al., 1995*). To investigate whether we could decode memory content for currently and prospectively relevant objects, we trained a classifier on the multivoxel response patterns in pFs using each variable template category (i.e., cow, dresser and skate) for each repetition time (TR). Here, we focus on the multivariate representation of the template categories of interest, as shown in *Figure 2*. The mean BOLD response for this area is shown in *Figure 2—figure supplement 1*. First, we trained and tested the classifier separately for trials in which the target category was currently relevant (for Search 1) and when the target category was prospectively relevant (for Search 2, see Methods section for details). Object category classification performance for this within-relevance decoding scheme is shown in *Figure 2A*. We focused our statistical analysis on the averaged classification performance for three intervals in the trial (of three TRs each; as predetermined on the basis of *Lee et al. (2013)*, referred to as Delay, Search 1, and Search 2; see Methods). We used paired t-tests (N = 24) to compare the classification performance to chance (33.33%) for these intervals, as well as between Current and Prospective conditions. *Figure 2B* shows the average activity for these time windows.

Second, while the within-relevance decoding scheme provides evidence for the presence of current and prospective representations, it does not reveal whether these representations are similar or different. Therefore, we additionally planned a cross-relevance decoding scheme in which we trained the classifier when the objects were currently relevant and tested when the same objects were prospectively relevant (referred to as PC), and vice versa (referred to as CP, see Methods). *Figure 2C and D* show the classification accuracy for this cross-relevance decoding scheme. Crucially, if current and prospective template representations are similar, above-chance classification accuracy is expected. If representations are dissimilar in an unrelated fashion, classification is expected to be at chance levels, while below-chance classification is predicted when representations are dissimilar, but in a systematic, anti-correlated fashion. Our general starting hypothesis was that while current and prospective representations would be similar during encoding, they would become increasingly dissimilar during the course of the trial, due to reduced activity or re-coding of the prospective item within the same network, while becoming similar again when the prospective memories are revived for the second task.

## The delay prior to the first search: Stronger decoding for current than for prospective templates, but similar representations

As can be seen in *Figure 2A and B*, during the Delay prior to search the within-relevance decoding resulted in significant above chance object category decoding both when the variable template was currently relevant ($t_{(1,23)}$ = 8.18, p < 0.001, $d$ = 1.67) and when prospectively relevant ($t_{(1,23)}$ = 5.67, p < 0.001, $d$ = 1.16). However, while decoding accuracy for the current representation remained significantly above chance up and beyond the Search 1 display, the prospective representation returned to baseline during the delay. Notably, decoding performance was higher when the item was currently relevant than when it was prospectively relevant (Current vs. Prospective: $t_{(1,23)}$ = 3.22, p = 0.004, $d$ = 0.66), consistent with its importance for the upcoming search task. Thus, object-selective cortex proves sensitive to object category as well as task-relevance prior to search.

Next, we used the *cross-relevance* decoding scheme to assess whether current and prospective targets shared the same neural representational pattern (see *Figure 2B and C*). This analysis revealed strong above-chance classification of the template category, regardless of the specific training scheme (PC: $t_{(1,23)}$ = 8.81, p < 0.001, $d$ = 1.80 or CP: $t_{(1,23)}$ = 9.04, p < 0.001, $d$ = 1.85). There was no difference in decoding performance between the two schemes (PC vs. CP: $t_{(1,23)}$ = 1.43, p = 0.167, $d$ = 0.29). These results indicate that during the delay prior to search, the representational pattern of the category was similar regardless of the current or prospective status of the object.

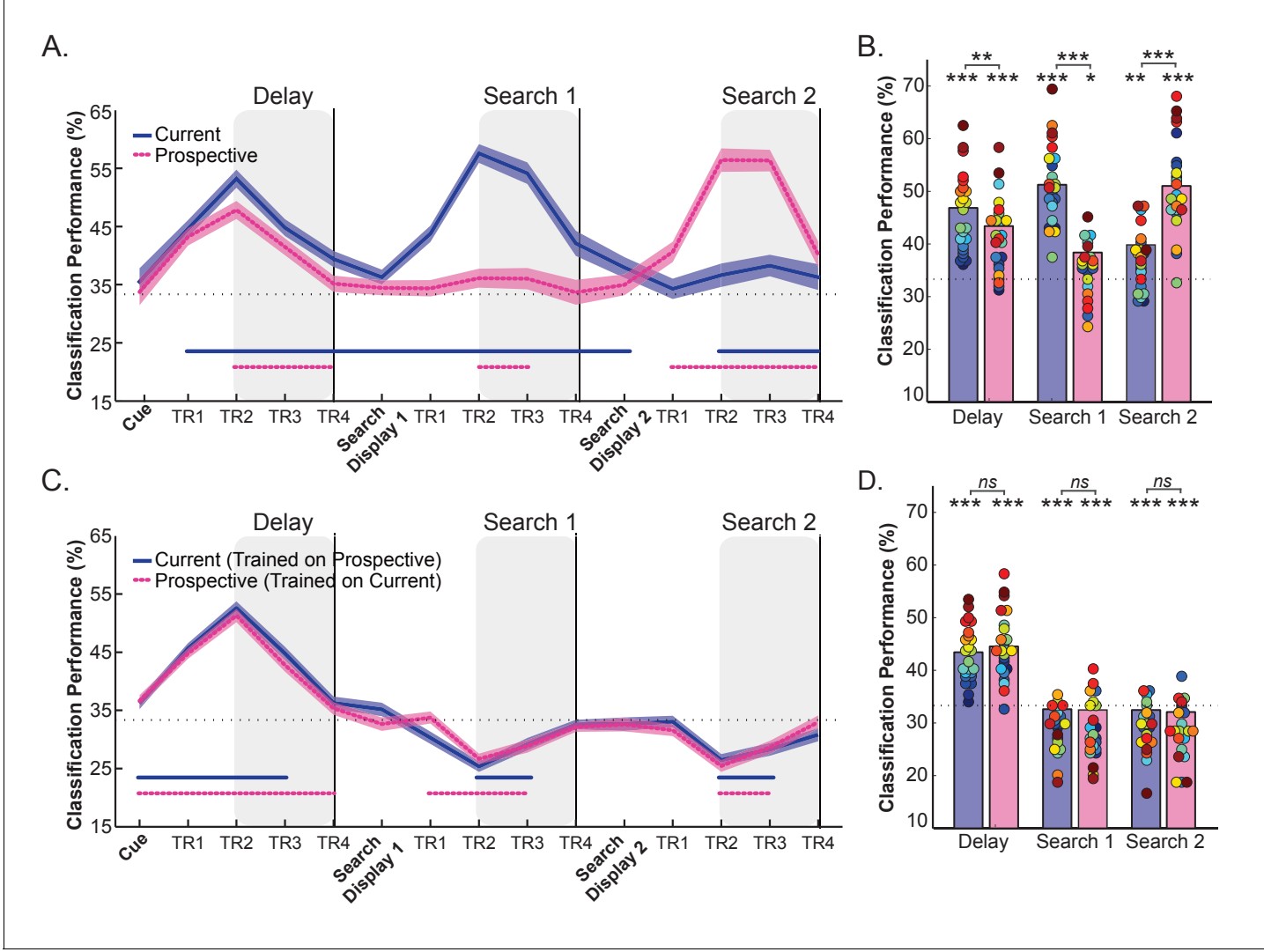

**Figure 2.** Within-relevance and Cross-relevance object category decoding in pFs. (A) Time course of the Within-relevance decoding where the classifier was trained and tested either within the current, or within the prospective conditions and (B) Average decoding accuracy within the time intervals shown by the shaded areas in (A). Decoding accuracy was higher for currently relevant templates (blue) than for prospectively relevant templates (pink) during the Delay and Search 1 intervals, and vice versa during the Search 2 interval. At the same time, the prospective template could still be reliably decoded during the first search, while the no longer relevant target could be decoded during the second search. (C) Time course of the Cross-relevance category decoding where the classifier was trained on current relevance, tested on prospective relevance, or vice versa and (D) Average decoding accuracy within the time intervals as shaded in (C). This resulted in above-chance decoding during the Delay prior to search (suggesting similar representations for current and prospective templates) but below-chance decoding during Search 1 and Search 2 (suggesting partially opposite representations). Shaded blue and pink areas indicate within-subjects s.e.m. Blue and pink horizontal lines at the bottom of the line graphs indicate time points that significantly differ from chance (p < 0.05, uncorrected). In the bar-plots, colored dots indicate individual participant data, N = 24. *p < 0.05, **p < 0.01, ***p < 0.001, ns: not significant.

DOI: https://doi.org/10.7554/eLife.38677.005

The following source data and figure supplements are available for figure 2:

**Source data 1.** Decoding performance for each participant of Experiment 1: includes source code and data to perform statistical analysis and produce *Figure 2*.

DOI: https://doi.org/10.7554/eLife.38677.008

**Source data 2.** Mean BOLD response for each participant of Experiment 1: includes source code and data to perform statistical analysis and produce *Figure 2—figure supplement 1*.

DOI: https://doi.org/10.7554/eLife.38677.009

**Source data 3.** Cross-temporal generalization matrices for each participant: includes source code and data to perform statistical analysis and produce *Figure 2—figure supplement 2*.

*Figure 2 continued on next page*

*Figure 2 continued*

DOI: https://doi.org/10.7554/eLife.38677.010

**Figure supplement 1.** Time course of the Mean BOLD response in area pFs for current and prospective trials of Experiment 1.

DOI: https://doi.org/10.7554/eLife.38677.006

**Figure supplement 2.** Cross-temporal generalization matrices for object category decoding as a function of Relevance.

DOI: https://doi.org/10.7554/eLife.38677.007

## Search 1: The prospective template can be decoded during the first search, but is different from its current counterpart.

Next, we wanted to know whether it was possible to successfully decode the category of the prospective template while participants were searching for a different object. As can be seen in *Figure 2A and B*, in the Search 1 interval we observed clear decoding of the object category when currently relevant (vs. 33.33%: $t_{(1,23)} = 11.57$, $p < 0.001$, $d = 2.36$), and this was stronger than when the object was prospectively relevant (between-condition comparison: $t_{(1,23)} = 9.42$, $p < 0.001$, $d = 1.92$). This is to be expected as during the first search of the Current condition (i.e., variable template search) the current template category is actually presented on the screen, whereas in the Prospective condition the objects on screen (i.e., flowers) differ from the prospective target in memory. Importantly, we were still able to also decode the prospective category during the first search (vs. 33.33%; $t_{(1,23)} = 1.90$, $p = 0.035$, $d = 0.39$). In other words, the prospective category re-emerges when observers are actively searching for an unrelated target.

This then raises the question as to whether the re-emerging prospective representation resembles its counterpart when currently relevant. To assess this we used the cross-relevance decoding scheme. Remarkably, here we observed *below*-chance decoding performance during Search 1 (CP: $t_{(1,23)} = -4.79$, $p < 0.001$, $d = -0.98$, and PC: $t_{(1,23)} = -3.67$, $p = 0.001$, $d = -0.75$). Although we hypothesized that current and prospective templates might differ in representational format, and thus cross-relevance decoding accuracy might have been reduced, we did not expect the sign of decoding to flip. The reliable deviation from chance further confirms that information on the prospective memory was present in object-selective cortex during the first search. In addition, the fact that decoding was below chance suggests that current and prospective representations of the same object category were represented through opposite multivariate patterns.

Finally, we assessed how the dissociation between current and prospective representations generalized across the different phases of the trial. As *Figure 2—figure supplement 2* shows, the pattern of activity prior to search is very similar to that during search for currently relevant representations, whereas prospectively relevant representations during the first search are markedly dissimilar from the same categories during the delay period prior to search. They then become similar again when retrieved for Search 2. Thus, while currently relevant representations remained constant from delay to search, the prospective representation was transformed from being similarly represented prior to search to being differently represented during search for the currently relevant item, followed by a reactivation for the second task.

## Search 2: Decoding of the first, now no longer relevant target during the second search.

Although not the primary goal of our study, we conducted the same analyses also for Search 2. As expected, here we saw the pattern reverse (see *Figure 2A and B*). In the within-relevance decoding scheme, we observed strong decoding of the category of the prospective target, which by now had become currently task-relevant (against chance, 33%: $t_{(1,23)} = 9.86$, $p < 0.001$, $d = 2.01$). This decoding was stronger than for the previously current search target, which was now no longer relevant ($t_{(1,23)} = -7.51$, $p < 0.001$, $d = -1.53$). Nevertheless, and unexpectedly, we also observed above-chance decoding for this first target during Search 2 ($t_{(1,23)} = 3.30$, $p = 0.002$, $d = 0.67$, blue). Note that this reflects classification of a target that is no *longer* relevant, whereas during Search 1 it reflected the target that was not *yet* relevant. Moreover, the cross-relevance decoding scheme also shows a pattern similar to what was observed during Search 1 (*Figure 2C and D*; and see also *Figure 2—figure supplement 2A* generalization across time). We found below-chance decoding in the same time range for both classification schemes (CP: $t_{(1,23)} = -4.65$, $p < 0.001$, $d = -0.95$ and PC:

$t_{(1,23)} = -4.49$, p < 0.001, $d = -0.92$). We will return to the re-emergence of the no longer relevant target later.

## Representational dissimilarity analyses comparing current and prospective representations

To further elucidate the relationship between current and prospective representations, *Figure 3A* shows, for each interval of interest (Delay, Search 1, Search 2), the representational dissimilarity matrices (*Kriegeskorte and Kievit, 2013*; *Kriegeskorte et al., 2008*). Specifically, we cross-correlated the voxel response patterns for every possible pair of the 24 stimulus/relevance combinations (4 exemplars x three categories (Cow, Dresser and Skate) × 2 relevance (Current and Prospective;

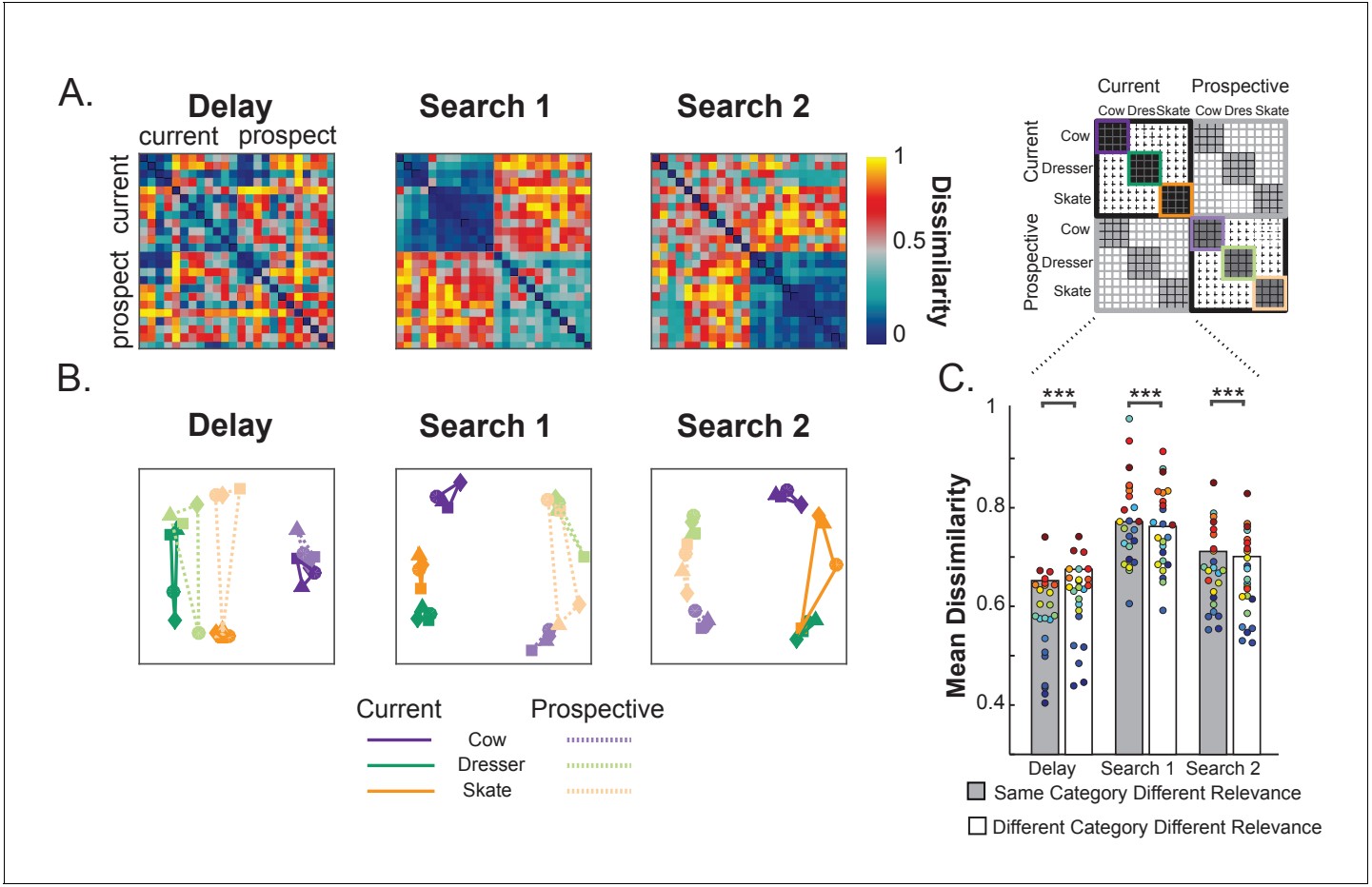

**Figure 3.** Representational dissimilarity analysis of object representations in pFs. (A) Representational dissimilarity matrices for the different variable template categories during Delay, Search 1 and Search 2, as a function of relevance (current and prospective). Blue indicates that representations are more similar while red indicates more dissimilar (B) Multidimensional scaling plots of the same similarity values, for the same Delay, Search 1 and Search 2 intervals. The four exemplars within each category are represented with different shapes (squares, triangles, circles and diamonds). The closer in space the more similar the neural representations. As a trial unfolds, object representations move from predominantly object category space during the delay prior to search (e.g. a cow) into predominantly relevance space (e.g. current target) during search, where current and prospective targets of the same category are represented by partly opposite representational patterns. (C) Comparing dissimilarity between Current and Prospective items when they are drawn from the same category versus when they are drawn from different categories. Representations prior to search within the same category are more similar than different categories, but this reverses during the searches. Colored dots indicate individual participant data, **p < 0.01, ***p < 0.001.

DOI: https://doi.org/10.7554/eLife.38677.011

The following source data is available for figure 3:

**Source data 1.** RDM for each participant of Experiment 1: includes source code and data to perform statistical analysis and produce *Figure 3*.
DOI: https://doi.org/10.7554/eLife.38677.012

see *Figure 3A* right panel and Methods for further details). To further visualize the relative position in multivariate space of each object category as a function of relevance, *Figure 3B* shows multidimensional scaling (MDS) graphs: the shorter the distance between categories the greater the representational similarity across voxels.

As can be seen from *Figure 3A and B*, throughout the course of the trial the neural representations moved from predominantly category space during the Delay period prior to search to predominantly relevance space during the two searches. Prior to search, objects grouped largely according to category, irrespective of relevance. This confirms that currently relevant and prospectively relevant objects were initially represented in similar ways in pFs. During Search 1, a clear relevance-driven distinction emerged between the neural object category representations. Note that this overall effect of relevance is probably driven by the fact that during search the currently relevant object category was presented in the display, while in the prospective condition the unrelated (flower) displays were presented. More interesting though is the finding that currently and prospectively relevant objects from the same category were represented as the most *dis*similar, as is illustrated by the representations taking opposite corners in the MDS plot in *Figure 3B*. For example, while all four exemplars of the cow category clustered together when all current, or all prospective, current cows were most separated from prospective cows – to the extent that current cows were more similar to prospective skates and dressers than they were to prospective cows. The same pattern held for the two other categories.

To statistically test these effects, we computed the average dissimilarity between current and prospective objects, separately for when drawn from the same category (e.g. current cow versus prospective cow) and when drawn from a different category (e.g. current cow versus prospective skate/dresser) and used paired t-tests (N = 24). *Figure 3C* shows these average same and different category dissimilarity values across relevance. During the Delay prior to the first search, as expected, *same* category representations were more similar than *different* category representations across relevance ($t_{(1,23)} = -5.82$, $p < 0.001$, $d = -1.28$, as performed on Fisher-transformed 1-rho values). In contrast, during Search 1, prospective targets differed most from current targets when they belonged to the same category, more so than when they belonged to different categories ($t_{(1,23)} = 3.06$, $p = 0.005$, $d = 0.64$). Likewise, during Search 2, no longer relevant targets were less similar from relevant targets when they belonged to the same category, than when they belonged to different categories ($t_{(1,23)} = 4.75$, $p < 0.001$, $d = 0.97$). Thus these analyses statistically confirm what we can observe from the MDS plots, namely that current and prospective objects move from similar to opposite representations.

## Experiment 2

What causes current and prospective representations to anti-correlate? One possibility is that the brain separates current from prospective templates within the same neuronal ensembles by actively transforming the representational pattern of prospective templates to be opposite to that of current templates. A second possibility is that the mechanism of making a representation prospective is more passive, and that the reversed pattern results from simply temporarily dropping a representation from memory, as it is temporarily irrelevant. One piece of evidence from Experiment 1 suggests that postponing and dropping an item may indeed involve similar mechanisms: During Search 2 we observed similar negative decoding for target representations that had served the first search and that were thus no longer necessary. However, getting rid of a competing target representation when switching to a new search may also involve active mechanisms, now to prevent interference from the past target rather than from a future target. Results from our lab indeed suggest such a suppression of previous targets (*de Vries et al., 2018*). A third possibility is that the pattern of reversal has little to do with memory whatsoever, but is simply a remnant of stimulus-related activity during encoding. Specifically, presenting the to-be-memorized stimulus may result in neural adaptation (*Henson and Rugg, 2003*; *Larsson and Smith, 2012*; *Vautin and Berkley, 1977*) or in a BOLD-related undershoot (*Huettel and McCarthy, 2000*), each of which would predict a reduced voxel response to later stimulation. Finally, the observed pattern of Experiment 1 may have been caused by certain idiosyncrasies of the experiment, most notably the fact that we always used the same flower target in the constant template condition, which may have led to either stimulus-specific or overall task difficulty-related interactions.

To address this, Experiment 2 sought to replicate and extend the main findings with a number of design changes. The most important change was the inclusion of a third condition, in which the memory item was cued to be irrelevant for any of the search tasks (see *Figure 1B* and Methods for details). Importantly, this Irrelevant condition matched the Prospective condition in all aspects – including the visual input – up to and including the first search, making the initial sensory processing identical across conditions. However, while in the Prospective condition the memory item was cued to become relevant only after the first search, in the Irrelevant condition the memory item was cued to become irrelevant altogether. Therefore, any differences in results across the Irrelevant and Prospective conditions can only be attributed to the future relevance of the memorized item and not to any systematic differences between memory items or search displays. For the same reason, neither can any differences between irrelevant and prospective representations be attributed to passive, sensory-related adaptation or BOLD undershoot.

Furthermore, we simplified the design by keeping the variable template search (here referred to as simply 'template search'), but replacing the constant template task of Experiment 1 with what we call a 'duplicate search' task, in which participants indicated whether or not one of the objects in the search display appeared twice (see *Figure 1B* and Methods for details, as well as *de Vries et al., 2018*). Note that such duplicate search does not require a template, because all the information needed to perform the task is in the search display itself. At the same time, it still engenders a dissociation between current and prospective memory templates over time. As a result, observers only had to remember a single target template per trial, which was either the target for the first search (Current; with the second search being a duplicate search), or the target for the second search (Prospective; with the first search being the duplicate search), or was deemed irrelevant after all (Irrelevant condition; with the first and only search being a duplicate search). Finally, just like the stimuli for the template search were drawn from three different categories (cows, skates, and dressers), we varied the stimuli in the duplicate search such that they were also drawn from three different categories (specifically butterflies, motorcycles, and trees), in order to assess whether the (below-chance) decoding of prospective representations during search generalizes across a range of different categories.

## Behavioral results

For both the template search and the duplicate search we computed mean RTs and mean percentage correct depending on the search order (see *Table 2*.). We ran a two-way repeated measures ANOVA (N = 25) with factors search order (Search 1 and Search 2) and type of task (template search and duplicate search), using data from the Current and Prospective conditions only (as the Irrelevant condition only had one search). The behavioral results show that performance in the template search and the duplicate search were comparable in terms of accuracy. There were no differences in accuracy between the two types of task as neither the main effects of search order ($F_{(1,24)}$ = 1.65, p = 0.210, $\eta_p^2$ = 0.065), type of task ($F_{(1,24)}$ = 2.48, p = 0.128, $\eta_p^2$ = 0.094), nor their interaction ($F_{(1,24)}$ = 0.16, p = 0.747, $\eta_p^2$ = 0.004) were significant. However, overall participants were faster in the first than in the second search task ($F_{(1,24)}$ = 37.24, p < 0.001, $\eta_p^2$ = 0.60) and faster in the template search than in the duplicate search ($F_{(1,24)}$ = 19.56, p < 0.001, $\eta_p^2$ = 0.44). There was also a significant

**Table 2.** Percentage correct and RT in Search 1 and Search 2 (N = 25) as a function of condition.

| | Current | | Prospective | | Irrelevant |
|---|---|---|---|---|---|
| | Search 1 | Search 2 | Search 1 | Search 2 | Search 1 |
| | Template | Duplicate | Duplicate | Template | Duplicate |
| P. Correct (%) | 86.0 (8.3) | 84.3 (6.9) | 83.9 (5.5) | 83.2 (8.4) | 83.4 (6.9) |
| RT (ms) | 1355 (96) | 1478 (114) | 1460 (90) | 1447(113) | 1469 (91) |

DOI: https://doi.org/10.7554/eLife.38677.013

The following source data is available for Table 2:

**Source data 1.** Behavioral data for each participant of Experiment 2.
DOI: https://doi.org/10.7554/eLife.38677.014

interaction between the two factors ($F_{(1,24)}$ = 14.76, p = 0.001, $\eta_p^2$ = 0.38), reflecting the fact that participants were faster in the template search when it occurred first (i.e. Current condition) than second (i.e. Prospective condition, $t_{(1,24)}$ = −7.40, p < 0.001, d = −1.48), while for the duplicate search, speed was the same regardless of the order ($t_{(1,24)}$ = −1.47, p = 0.154, d = −0.294).

The fact that participants were equally accurate at finding the template object regardless of the search order and that they were slower in the template search when performed second (i.e., Prospective condition) suggests that the quality of the memory representation did not decay when it had to be postponed, but reactivating it for Search 2 required additional time. So any differences between current and prospective representations were not simply due to participants being worse on the prospective memory. Moreover, while in Experiment 1 the search of interest (variable template) was more difficult than the remaining search task (constant template), here, if anything, it was the other way around. Yet, the fMRI findings show a pattern very similar to that of Experiment 1, as reported next.

## fMRI results: Target category decoding as a function of task relevance

As in Experiment 1, here we focus on multivariate representational patterns within the same three intervals: Delay, Search 1, and Search 2 (see *Figure 4* and Methods for details), here for Current, Prospective, and Irrelevant objects. The mean overall BOLD response in pFS is shown in *Figure 4—figure supplement 1*.

## The delay prior to the first search: Stronger decoding for current and prospective templates than for irrelevant items, but similar representations across conditions.

*Figure 4A and B* show the decoding accuracy during the Delay prior to search. A one-way ANOVA on the within-relevance decoding of Current, Prospective and Irrelevant conditions revealed a significant effect of condition ($F_{(2,48)}$ = 4.40, p = 0.018, $\eta_p^2$ = 0.15). As shown in *Figure 4B*, decoding accuracy was highest for Current, lowest for Irrelevant, with the Prospective condition in between. There was significant above-chance object category decoding for all relevance conditions: Current ($t_{(1,24)}$ = 10.76, p < 0.001, d = 2.15) Prospective ($t_{(1,24)}$ = 5.88, p < 0.001, d = 1.17) and Irrelevant ($t_{(1,24)}$ = 5.61, p < 0.001, d = 1.12). Pairwise comparisons revealed the difference between Current and Irrelevant to be significant ($t_{(1,24)}$ = 3.74, p = 0.001, d = 0.74). In contrast to Experiment 1 though, the difference in decoding accuracy between the Current and Prospective conditions was not significant ($t_{(1,24)}$ = 1.33, p = 0.193, d = 0.26), nor was there a significant difference between the Prospective and Irrelevant conditions ($t_{(1,24)}$ = 1.39, p = 0.175, d = 0.27). We will return to the possible reasons for this difference between experiments in the Discussion.

Next, we used the *cross-relevance* decoding scheme to assess whether current, prospective and irrelevant items shared the same neural representational pattern (see *Figure 4C and D*). The classification performance of each particular train-test scheme and its converse counterpart were averaged (see Methods for details). This analysis revealed above-chance classification for the three cross-relevance decoding schemes: Current-Prospective ($t_{(1,24)}$ = 9.85, p < 0.001, d = 1.97), Current-Irrelevant ($t_{(1,24)}$ = 8.10, p < 0.001, d = 1.62) and Prospective-Irrelevant ($t_{(1,24)}$ = 6.90, p < 0.001, d = 1.38). A one-way repeated measures ANOVA with decoding scheme as factor revealed no significant differences between decoding schemes ($F_{(2,48)}$ = 2.4, p = 0.101, $\eta_p^2$ = 0.09). Taken together, these results indicate that during the delay prior to search, the representational pattern of the memorized object category was similar regardless of the current, prospective or irrelevant status of the object.

## Search 1: The prospective target but not the irrelevant item can be decoded during the first search

A one-way ANOVA on the within-relevance decoding during Search 1 showed a significant difference in decoding accuracy ($F_{(1.6,38.77)}$ = 35.17 p < 0.001, $\eta_p^2$ = 0.59, corrected for sphericity). Decoding accuracy for the current template was above chance ($t_{(1,24)}$ = 9.47, p < 0.001, d = 1.89) and significantly higher than for both the prospective template ($t_{(1,24)}$ = 5.53, p < 0.001, d = 1.10) and the irrelevant item ($t_{(1,24)}$ = 7.29, p < 0.001, d = 1.45). This was to be expected since during the first search of the Current condition, the current template category was on the screen. More importantly,

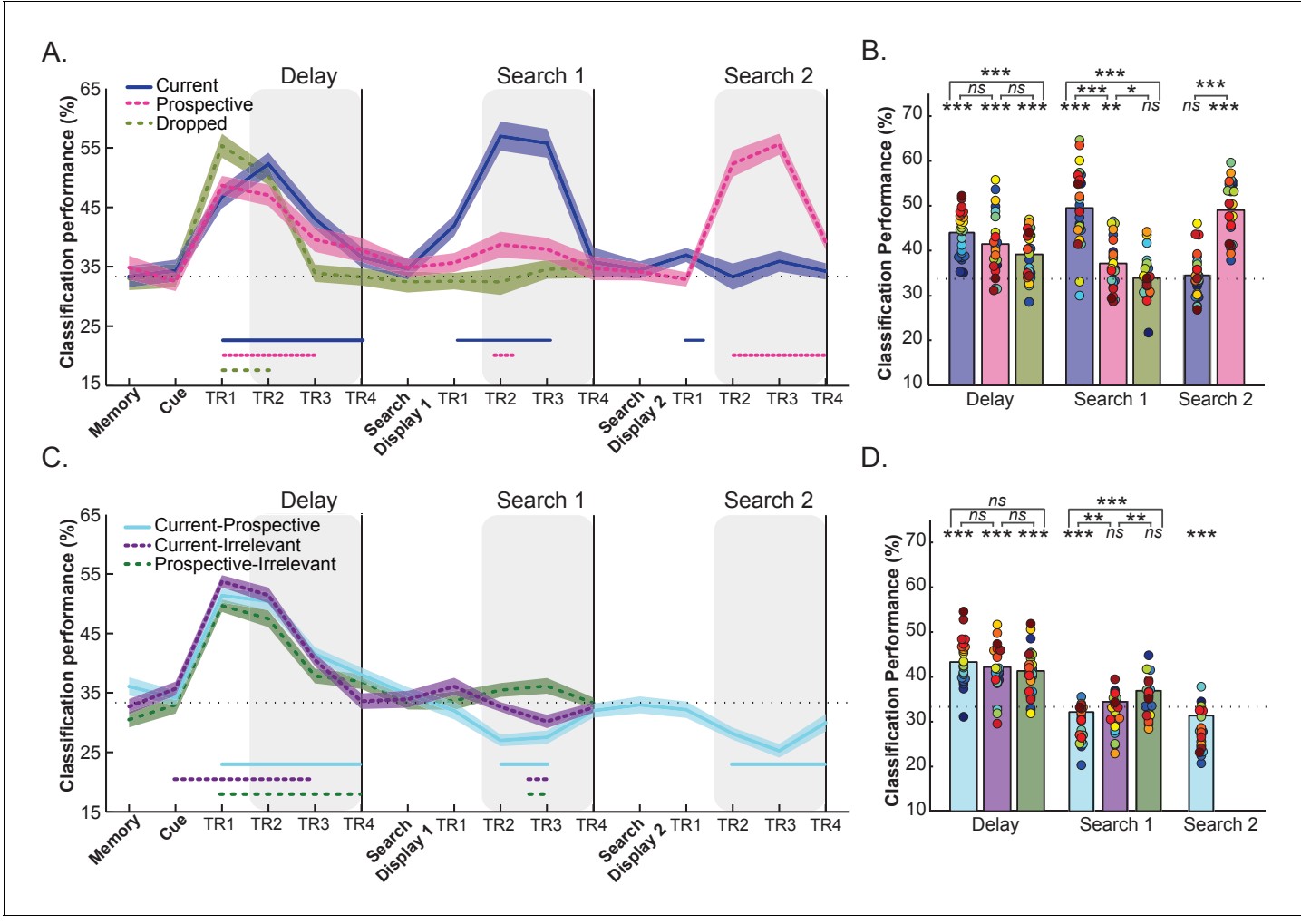

**Figure 4.** Within-relevance and cross-relevance object category decoding in pFs. (**A**) Time course of within-relevance decoding and (**B**) Averaged decoding accuracy within the time intervals shown by the shaded areas in A. During the delay, object category decoding was higher for currently relevant objects (blue) than for irrelevant objects (green) with in between decoding accuracy for prospective templates. During Search 1 the current template showed higher decoding accuracy than the prospective template and the irrelevant item. Importantly, the category of the prospective template could also be decoded during the first search, while the irrelevant category was at chance. During Search 2 the prospective (now current) category was clearly decodable while the formerly current (now no longer relevant) category was at chance (**C**) Time course of cross-relevance decoding and (**D**) Averaged decoding accuracy within the time intervals shown by the shaded areas in C. Classification was above chance for all decoding combinations during the Delay prior to search, suggesting similar representations for current, prospective and irrelevant objects. In contrast, we observed below-chance decoding during Search 1 and Search 2 for the Current-Prospective (blue) cross-relevance scheme (suggesting systematically different representations); importantly, this was stronger than for Current-Irrelevant (purple; during Search 1). Current-Irrelevant (purple) and Prospective-Irrelevant (green) cross-classification schemes were at chance. Shaded areas indicate within-subjects s.e.m. Blue and pink, purple and green horizontal lines at the bottom of the line graphs indicate time points that significantly differ from chance (p < 0.05, uncorrected). In the bar-plots, colored dots indicate individual participant data, N = 25. *p < 0.05, **p < 0.01, ***p < 0.001, ns: not significant.
DOI: https://doi.org/10.7554/eLife.38677.015

The following source data and figure supplement are available for figure 4:

**Source data 1.** Decoding performance for each participant of Experiment 2: includes source code and data to perform statistical analysis and produce *Figure 4*.
DOI: https://doi.org/10.7554/eLife.38677.017

**Source data 2.** Mean BOLD response for each participant of Experiment 2: includes source code and data to perform statistical analysis and produce *Figure 4—figure supplement 1*.
DOI: https://doi.org/10.7554/eLife.38677.018

**Figure supplement 1.** Time course of the Mean BOLD response in area pFs of Experiment 2.
DOI: https://doi.org/10.7554/eLife.38677.016

and in line with Experiment 1, we were able to decode the prospective category at above-chance levels during Search 1 ($t_{(1,24)}$ = 3.35, p = 0.003, d = 0.67). At the same time, decoding of the *Irrelevant* category remained at chance ($t_{(1,24)}$ = 0.60, p = 0.550, d = 0.12), and was significantly weaker than for Prospective condition ($t_{(1,24)}$ = 2.27, p = 0.032, d = 0.45). Thus, information about the prospective template can be recovered while participants perform a different search task, whereas completely irrelevant categories are no longer decodable.

Next, we used the cross-relevance decoding scheme to evaluate whether the representations of the prospective targets and irrelevant items resemble their current counterpart (see *Figure 4C and D*). A one-way ANOVA with decoding scheme (Current-Prospective, Current-Irrelevant, and Prospective-Irrelevant) revealed a reliable effect ($F_{(2,48)}$ = 16.35, p < 0.001, $\eta_p^2$ = 0.40). Replicating Experiment 1, we observed strong *below*-chance decoding during Search 1 for the Current-Prospective cross-relevance classification ($t_{(1, 24)}$=−5.92, p < 0.001, d = −1.18), while the Current-Irrelevant ($t_{(1, 24)}$= −1.89, p = 0.071, d = −0.37) and the Prospective-Irrelevant ($t_{(1,24)}$ = 1.99, p = 0.058, d = 0.39) cross-decoding schemes did not significantly differ from chance. Most importantly, decoding accuracy for the Current-Prospective scheme was significantly *lower* than for the Current-Irrelevant scheme ($t_{(1,24)}$ = −2.95, p = 0.007, d = −0.59), indicating that the Prospective condition involved a stronger representational transformation than the Irrelevant condition. Since the prospective and irrelevant trials contained exactly the same visual input (see Methods), this result suggests that this transformation is driven by the future relevance of the prospective template rather than by some passive, automatic mechanism that would be the same for prospective and irrelevant representations (such as BOLD undershoot, see Discussion).

## Search 2: Evidence for the first, no-longer relevant target during the second search

In line with Experiment 1, the pattern reversed for Search 2 (see *Figure 4*). In the within-relevance decoding scheme, we observed strong decoding of the category of the prospectively relevant target, which by now had become task-relevant ($t_{(1,24)}$ = 12.54, p < 0.001, d = 2.50), and decoding of the Prospective – now relevant - category was stronger than for the previously *current* – now no longer relevant – search target ($t_{(1,23)}$ = −8.84, p < 0.001, d = −1.76). In contrast to Experiment 1, where we unexpectedly observed above-chance decoding for this no longer relevant target during Search 2, here the current condition was at chance ($t_{(1,24)}$ = 1.14, p = 0.265, d = 0.22). However, similar to Experiment 1, we found below-chance decoding in the same time range for the Current-Prospective cross-relevance decoding scheme (vs. 33.3%: $t_{(1,24)}$ = −6.57 p < 0.001, d = −1.31), indicating that there was information present on the previously relevant target.

## Representational dissimilarity analyses comparing current, prospective and irrelevant representations

Next, we created representational dissimilarity matrices and multidimensional scaling (MDS) graphs for each interval of interest (Delay, Search 1, Search 2) for the three possible condition combinations: Current-Prospective, Current-Irrelevant and Prospective-Irrelevant (see *Figure 5*). First, as can be seen in *Figure 5A, B and C*, we replicated the results from Experiment 1. In the Delay period, the neural representations of current and prospective targets of the *same* category were more similar (i. e., less dissimilarity) than the representations of targets from *different* categories ($t_{(1,24)}$ = −9.14, p < 0.001, d = −1.82, *Figure 5C*). In contrast, during Search 1 and Search 2, currently and prospectively relevant objects from the same category were more *dis*similar: As in Experiment 1, prospective targets differed most from current targets when they belonged to the *same* category than when they belonged to *different* categories (Search 1: $t_{(1,24)}$ = 5.20, p < 0.001, d = 1.04 and Search 2: $t_{(1,24)}$ = 6.45, p < 0.001, d = 1.29). Again, throughout the course of the trial, the neural representations of the target objects moved from predominantly category space in the Delay period prior to search to predominantly relevance space during the two searches, where current and prospective objects were represented in opposite corners of the representational space.

The results for the Current-Irrelevant (*Figure 5D, E and F*) and Prospective-Irrelevant (*Figure 5G, H and I*) condition combinations were very comparable during the delay prior to the first search. Here too, the neural representation of targets of the same category were more similar than when they were from different categories (Current-Irrelevant: $t_{(1,24)}$ = −8.27, p < 0.001, d = −1.65 and

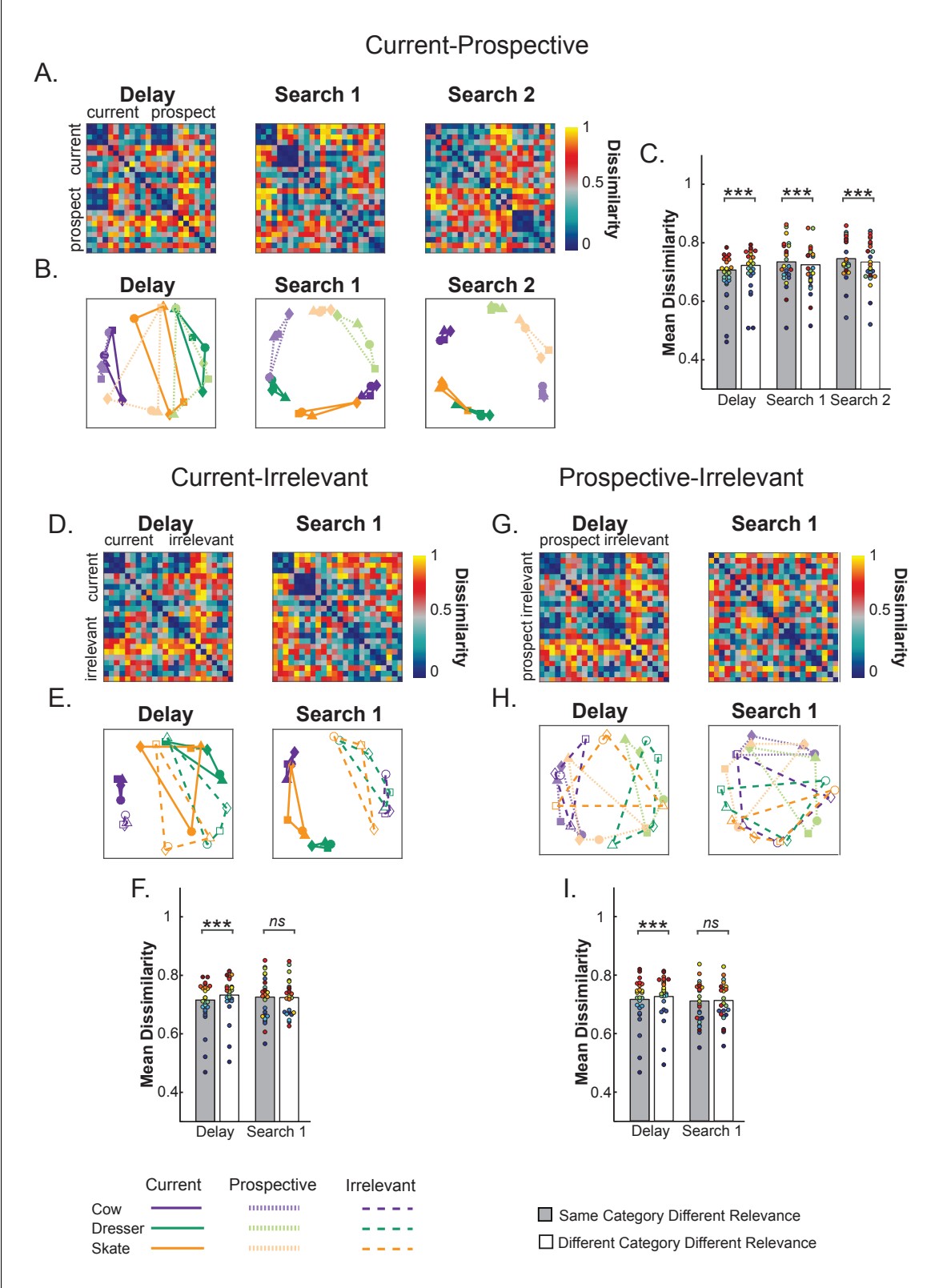

**Figure 5.** Representational dissimilarity analysis of object representations in pFs. (A,D,G) Representational dissimilarity matrices and (B,E,H) Multidimensional scaling plots of the same similarity values for the different target object categories during Delay, Search 1 and Search 2, as a function of relevance. The four exemplars within each category are represented with different shapes (squares, triangles, circles and diamonds). (A,B) Comparing Current to Prospective: Object representations moved from predominantly object category space (e.g. a cow) during the Delay period to

*Figure 5 continued on next page*

*Figure 5 continued*

predominantly relevance space during search, where current and prospective targets of the same category were represented by partly opposite representational patterns. (C) Current-Prospective Mean Dissimilarity. During the delay prior to search, representations within the same category are more similar than of different categories; however, the pattern reverses during Search 1 and Search 2, where representations within the same category are more dissimilar than of different categories. (D, E, F) Comparing Current to Irrelevant and (G, H, I) Comparing Prospective to Irrelevant. During the delay prior to search, targets of the same categories were more similar than of different categories, with no opposite representational pattern during search. Colored dots indicate individual participant data, **p < 0.01, ***p < 0.001, ns: not significant.

DOI: https://doi.org/10.7554/eLife.38677.019

The following source data is available for figure 5:

**Source data 1.** RDM for each participant of Experiment 2: includes source code and data to perform statistical analysis and produce *Figure 5*.
DOI: https://doi.org/10.7554/eLife.38677.020

Prospective-Irrelevant: $t_{(1,24)}$ = −4.73, p = 0.001, $d$ = −0.94). Importantly however, and in contrast to Prospective items, for Irrelevant items we did not find evidence that the representations were warped in opposite corners of the multivariate space during Search 1, as the (dis)similarity between targets belonging to the same or to different categories was equal for the Current-Irrelevant ($t_{(1,24)}$ = 0.94, p = 0.354, $d$ = 0.89) and Prospective-Irrelevant ($t_{(1,24)}$ = −1.36, p = 0.184, $d$ = −0.27) comparisons.

The uniqueness of the prospective transformation was further confirmed by directly comparing the relative similarity difference (by subtracting the mean target dissimilarity when of the *same* category from the mean dissimilarity when of *different* category) for each relevance combination (i.e., Current-Prospective, Current-Irrelevant and Prospective-Irrelevant). This showed that same category representations were indeed relatively more dissimilar for Current-Prospective comparisons than for Current-Irrelevant ($t_{(1,24)}$ = 2.88, p = 0.008, $d$ = 0.57) and Prospective-Irrelevant ($t_{(1,24)}$ = 4.69, p < 0.001, $d$ = 0.93) comparisons.

Taken together, the results of Experiment 2 confirm the most important findings of Experiment 1. We find a pronounced anti-correlated representation for prospective targets compared to the same targets when current. Experiment 2 furthermore shows that it is not the temporary irrelevance, but the prospective relevance that causes this transformation, because little to no anti-correlation was found for items that were irrelevant altogether. Since prospective and irrelevant trials contained exactly the same visual input (up to Search 1 inclusive), the transformation of prospective representations cannot be the result of basic stimulus-induced neural adaptation or BOLD response properties.

## Discussion

It has been shown that the content of working memory can be decoded from multivariate patterns of voxel activity when observers are remembering an item for a single task (*Albers et al., 2013*; *Harrison and Tong, 2009*; *Lewis-Peacock and Postle, 2012*; *Serences et al., 2009*). Furthermore, working memory representations have been shown to adapt to the specific task goal, as the representation of the same object changes depending on the nature of the upcoming test (*Lee et al., 2013*; *Myers et al., 2017*). Here, we provide further evidence for the flexibility of working memory by showing how it not only distinguishes between relevant and irrelevant representations, but also between relevant representations for current and future task goals, as representations adapt to the order in which they are required in multiple task sequences.

In line with earlier work in different task and stimulus domains (*LaRocque et al., 2013*; *LaRocque et al., 2017*; *Lewis-Peacock and Postle, 2012*; *Wolff et al., 2017*), we observed that objects required for an upcoming search task are represented more strongly than when the same objects are only used prospectively (Experiment 1) or when they are completely irrelevant (Experiment 2), as was indicated by stronger classification performance in object-selective cortex prior to the first search. Furthermore, corroborating findings by *Rose et al. (2016)*, we observed that the prospectively relevant memory could be reconstructed during task-irrelevant stimulation, here during the first search for an unrelated stimulus. This was shown in two ways: First, both experiments showed above-chance classification of the prospective item during the first search, whereas Experiment 2 showed that classifier performance for irrelevant representations remained at chance and was significantly worse than for prospective representations. Second, using a cross-relevance training

scheme, where the classifier was trained when the item was current and tested when it was prospective (or vice versa), we also found in both experiments that decoding performance reliably differed from chance – but now in a negative direction.

Our results are the first to reveal a direct relationship between currently relevant object representations on the one hand, and prospective representations on the other. Prior to the first search current, prospective and irrelevant representations were very similar, as cross-relevance decoding (training on one status while testing on the other) showed above-chance performance for all classification schemes and there were no differences across conditions. However, while the representation for prospective objects clearly reversed during search, such a reversal was weak to absent for irrelevant objects. We observed that during search, the reconstructed prospective memory representations of objects proved *dis*similar from their current counterparts. Importantly, they differed in a systematic manner, to the extent that prospective representations were even more dissimilar from current representations of the *same* object category than representations of a *different* object category, and were characterized by an inverse correlation with current representations. Conversely, the initial similarity between current and irrelevant targets, which was evident during the delay, mostly disappeared during search leading to the same dissimilarity values for targets of the same vs. different category. Thus, while irrelevant targets simply appeared to decay from memory, prospective target memories were transformed. We point out that similar (as yet unpublished) findings have recently been reported by *Yu and Postle (2018)*, for a different stimulus class and a different brain area. They asked participants to memorize two oriented gratings, one for a first test (making it current), the other for a second test (making it prospective). Using multivariate inverted encoding models of occipital cortex activity, they could reconstruct prospectively relevant orientations with models trained on currently relevant orientations, but here too a reversed pattern emerged. Together with our findings, these results indicate that prospective targets may be dissociated from current targets in two ways. First, they appear distinct in that current representations are activity-based, whereas prospective representations are responsivity-based. Second, prospective targets are represented through a responsivity pattern different to that of current target activity, where the most active part becomes the relatively least responsive and vice versa.

An issue left unresolved in the present study is whether the observed transformation of prospective representations extends to specific exemplars. That is, are current and prospective representations even more dissimilar when representing the same exemplar? Unfortunately, the limited number of trials per exemplar and condition precluded a useful analysis here, and future studies should directly test the item-specificity of the transformation of prospective representations.

Imaging studies of visual attention have demonstrated that assigning attentional priority to task-relevant objects at the expense of irrelevant objects leads to the transformation of the representational space, by relatively enhancing target-related distributed activity (*Reddy et al., 2009*) and by recruiting additional resources as neurons shift their tuning towards the attended object category (*Çukur et al., 2013*; *Nastase et al., 2017*). Here, such relative enhancement can explain the pattern of results during the delay period prior to the first search task in Experiment 1, where we found a difference in representational strength between current and prospective templates, while their representations remained similar. Note that no such difference occurred in the delay period of Experiment 2. This discrepancy between experiments may be explained by assuming a direct competition between current and prospective templates, which was only present in Experiment 1 (where there was a variable and a constant template). In Experiment 2 there was only one object to remember, allowing observers to devote the same resources in all conditions.

A crucial question that our data do not answer is what the exact mechanism is behind the transformation from current to prospective representations. One possibility is the involvement of an active cognitive control mechanism, which specifically attempts to dissociate prospectively relevant from currently relevant representations in order to prevent task interference. Such control mechanisms might be exerted through feedback connections emanating from frontal areas central to counteracting unwanted or task-irrelevant information (*Anderson et al., 2004*; *Banich et al., 2015*; *de Vries et al., 2018*; *Depue et al., 2007*; *Reeder et al., 2017*). Interestingly, an earlier study of memory retrieval has shown suppression of voxel patterns in ventral object-related cortex which were associated with task-irrelevant memories of learned object pictures, leading to comparable patterns of representational dissimilarity as here (*Wimber et al., 2015*). Initial evidence for the suppression of temporarily irrelevant items also comes from a study by *Peters et al. (2012)*, who used a

similar task design as ours. They asked observers to consecutively look for a particular house target and a particular face target (or vice versa) in rapid streams of house/face distractors. They found the overall BOLD signal to be reduced in either house or face selective areas in response to house/face stimuli when the respective target was prospectively relevant. Here, we show how changing task relevance within working memory specifically affects the cortical pattern of activation within memory while observers perform a different search task. In this respect it is also interesting to note that in both experiments we found evidence for an inversion both for temporarily irrelevant targets (i.e. prospective targets during the first search), and targets that were no longer relevant (i.e. previous targets during the second search). This suggests a shared mechanism for preventing interference, whether from future targets or from past targets. The results of Experiment 2 would then imply that items that were never a template for search in the first place, on a certain trial (i.e. the irrelevant condition), would interfere less with the subsequent task and thus did not need to be transformed.

An alternative possibility is that local, and arguably more passive mechanisms cause the change in responsiveness. We believe that we can exclude at least two of such mechanisms on the basis of the current data, specifically sensory-induced neural adaptation, and at a macro level, a sensory-induced BOLD undershoot. Both mechanisms would predict the voxels that were most active during memory encoding to be least active a short while later. In fact, such adaptation in responsivity might be functional in memory retrieval (*Meyer and Rust, 2018*; *Turk-Browne et al., 2006*; *Ward et al., 2013*), and may even also explain the earlier demonstrations of memory reconstruction by *Rose et al. (2016)* and *Wolff et al. (2017)*. However, Experiment 2 here showed a differential neural response for prospective and irrelevant items, despite the fact that stimulus presentation was identical. Thus, the representational differences we find are determined by the observer's goal state, and not automatically induced by the sensory representation of the stimulus.

Although the underlying mechanism remains unknown, we believe the results have important implications for theories of prospective memory storage in working memory. First, the fact that prospectively relevant objects could be decoded from the same regions of interest as the currently relevant objects indicates that the different memory states do not necessarily rely on different brain areas. The successful cross-relevance decoding, where we trained the classifier on one state and tested on another, further confirms this. Second, the idea that prospectively relevant memories are stored in an activity-silent format has recently been debated by *Schneegans and Bays (2017)* on the basis of the argument that existing data can also be explained by a simpler model which assumes that temporarily irrelevant memories are represented through the same activity as relevant memories, but in a weaker form. *Schneegans and Bays (2017)* argued specifically against a study by *Sprague et al. (2016)*, which indeed showed clear remnants of activity for representations that were assumed to be partly latent. But even when the data reveal no such activity this may only reflect the limited sensitivity of the measure at hand. The reduced activity account is partially supported by our data. We found that during the delay period prior to the first search task, before the evidence for the prospective item diminished to baseline levels, current and prospective representations were highly similar, as evidenced by a strong correlation and successful cross-relevance classification of the current, prospective and irrelevant representations. However, the reduced activity account does not explain that current and prospective representational patterns were very dissimilar during the search. In fact, the partial anti-correlation indicates suppression rather than activation of the relevant voxels.

Instead, the emergence of the prospective memory that we found here during the first search fits best with a change in responsivity, resulting in an activity-silent representation. The fact that it was necessary to add activity to the system for the prospective memory to emerge – here in the form of unrelated visual search displays, is already testament to this. Importantly, the current data put limits on the potential mechanisms by which the responsivity changes. A frequent hypothesis is that prospectively relevant representations are stored through temporary synaptic potentiation. Such short-term potentiation predicts that what was strongly activated during encoding, will become more responsive, when prospective, and fire more strongly when reactivated. We observed the opposite: What was strongly activated when current became more strongly suppressed when prospective and vice versa. This goes against a simple short-term potentiation account of activity-silent representations in working memory.

In conclusion, we find evidence that, in trying to separate current and prospective goals in visual search, the brain stores representations within the same neuronal ensembles, but through opposite representational patterns.

# Materials and methods

## Participants

Twenty-four participants (eight males, M = 26.74 years of age, SD = 3.21 years) participated in Experiment 1, and twenty-five (based on Experiment 1, we planned 24 participants. We tested an extra subject to ensure that we would have a complete sample in case a participant had to be excluded. In the end, this turned out unnecessary, and all tested participants were included in the analyses) participants in Experiment 2 (14 males, M = 25 years of age, SD = 4.5 years). For both experiments, we obtained written informed consent from each participant before experimentation. Participants had normal or corrected-to-normal vision. The experiment was approved by the Ethical Committee of the Faculty of Social and Behavioral Sciences, University of Amsterdam (where scanning took place) and conformed to the Declaration of Helsinki.

## Task and stimuli

### Experiment 1

On each trial, participants performed two consecutive visual search tasks of real-world objects. The object of interest (cow, dresser or skate) consisted of real-world greyscale photographs, selected out of four exemplars. These categories were selected to have maximal dissimilarity in representational space (see *Harel et al., 2014*). The object of interest (cow, dresser or skate) was to be searched for first or second – thus making it currently or prospectively relevant (referred to as the variable template search). To maximize the chances of decoding the target of interest (whether current or prospective), and to limit the working memory load, the remaining search task always involved the same 'daisy' flower target (i.e. Only one exemplar) referred to as the constant template search.

As can be seen in *Figure 1*, each trial started with a fixation followed by the sequential presentation of two memory items (variable template [cow, dresser or skate] and constant template [the daisy], 2.4° visual angle) each presented for 750 ms with a 500 ms fixation in between. After a fixation of 500 ms a cue, either a 1 or a 2 was presented indicating the search order in which the memory items needed to be searched for in two subsequent search tasks. Thus, participants either had to search for the variable template first and then the constant daisy template (Current condition) or the daisy had to be searched for first and the variable template second (Prospective condition). Both relevance conditions (Current and Prospective), order of the memory items as well as the cue were counterbalanced across trials. The cue was followed by an 8 s delay with a fixation dot in the middle of the screen ('Delay') after which the first search display was presented. The search display consisted of six different exemplars (2.4° visual angle) of the same category as the cued memory item and could either contain the remembered 'Current' object ('Present') or not ('Absent'). Participants had to indicate through button presses with their left and right hand whether the memory item was present or absent. The distractors in the search displays were randomly placed among a radius of (7.4° visual angle). The search display was presented for two seconds and participants had to respond within these two seconds. After the first search display another eight seconds blank delay period followed ('Search 1') and then the second search display was shown depicting exemplars from the uncued object category. This was again followed by an eight second inter trial interval (ITI) ('Search 2'). After completion of the first search task, observers had to turn to the prospective item, and indicate its presence or absence in the second search display. At the end of each trial, within the ITI, participants received feedback (for 400 ms) on their performance for each search tasks: either 'correct', 'incorrect' or 'missed' (if the response did not occur within the 2 s duration of the search displays). At the end of each run the percentage correct and average reaction times were presented for both the constant template search and the variable template search.

Notice that in Experiment 1, the current and prospective templates were always drawn from separate category sets within a trial. Specifically, in current trials, the current item was drawn from one of the three categories in the variable template set (object of interest: cows, dressers or skates) and

the prospective item was always the same constant template flower. In prospective trials the category sets reversed, with the prospective item being the variable template. We did this intentionally; having independent sets for the two search templates - within a trial - ensured that we could unequivocally interpret the category classification accuracy as reflecting the representation of the object of interest when either current or prospective. Remember that the classifiers learned to differentiate the neural pattern of the categories of interest: cow, dresser and skate. If both templates were to be drawn from the same category set within the trial, it would be impossible to know whether category classification accuracy actually reflects the quality of the representation of the current template, the prospective template or a combination of the two.

The main experiment consisted of 8 runs with 12 trials each (96 trials in total). Each run contained equal amount of trials per condition and category of interest (i.e., cow, dresser and skate). Each experimental run lasted ~7 min. The total duration of a session was ~1.5 hr (including the structural scan (6 min) and mapper run (7 min), see below).

### Experiment 2

The task in Experiment 2 was similar to Experiment 1, but we included important changes (see *Figure 1B*). First, we replaced the constant template search with a duplicate search where participants had to indicate if one of the objects appeared twice in the search display. Second, the duplicate search tasks changed the category from trial to trial to be one out of three possible categories (butterfly, motorcycles and trees). Finally, we added a third condition (i.e., Irrelevant condition), where after the cue participants could immediately drop the item from memory as they only performed the duplicate search task.

Each trial started with the presentation of only one memory item (cow, dresser or skate) for 1500 ms, followed by a fixation display that stayed on for 1500 ms. Then, a cue indicated the relevance of the memory item. The cue could be either a 1, 2 or 0 and remained on the screen for 1000 ms. When the cue was '1', participants performed the template search first and the duplicate search second, making the memorized object currently relevant (Current condition). The order reversed when the cue was '2', rendering the memorized object only prospectively relevant, as observers performed the duplicate search first and the template search second (Prospective condition). Finally, if the cue was '0' the object was irrelevant because participants would only perform the duplicate search (Irrelevant condition) and would not be tested on the memory object. As in Experiment 1, the cue was followed by an 8 s delay with a fixation cross in the middle of the screen ('Delay') after which the first search display was presented. Depending on the condition, the first search display was either a template search (Current condition) or a duplicate search (Prospective and Irrelevant conditions). In the template search, participants indicated with a button press whether the memorized object of interest was present or absent among six exemplars of the same object category. Similarly, in the duplicate search, participants indicated whether a duplicate object (i.e., the same exemplar appeared twice in the search display) was present or absent, again set size for this display was six objects. After the first search display another eight seconds blank delay period followed ('Search 1') after which the trial either ended (Irrelevant condition) or the second search display was presented (Current and Prospective conditions). This second search display was also followed by an eight second blank period ('Search 2') after which the trial ended. The location of items and the duration of the search displays were the same as in Experiment 1.

The main experiment consisted of 9 runs with 12 trials each (108 trials in total). Within each experimental run, we balanced the amount of times that each relevance condition (Current, Prospective and irrelevant) was presented (four trials per condition), as well as the amount of times that participants had to respond either 'present' or 'absent' in each search task. However, the relevance condition by category combinations (i.e., nine in total: three relevance conditions [current, prospective, irrelevant] x three memory category [cow, dresser, skate]) could not be completely balanced within runs (12 trials per run); nonetheless, across the whole experiment there were equal amount of trials for each combination. We also balanced the category used in the duplicate search task (butterfly, motorcycles and trees) across conditions and in combination with the category of the variable template (i.e. cow, dresser, skate). Each experimental run lasted ~7 min. The total duration of a session was ~1.7 hr (including, short brakes in between runs, the structural scan and mapper run).

In both Experiments, the stimuli were back-projected on a 61 × 36 cm LCD screen (1920 x 1080 pixels) using Presentation (Neurobehavioral Systems, Albany, CA, USA) and viewed through a mirror attached to the head coil. Eye tracking data (EyeLink 1000, SR Research, Canada) was monitored to ensure participants were awake and attending the stimuli.

## Regions of interest: object-selective cortex mapper (pFs)

At the end of each session we independently mapped the region of interest as the region that responded more strongly to intact vs. scrambled objects (*Malach et al., 1995*), within an anatomical mask of the temporal occipital fusiform cortex (from the Harvard-Oxford Cortical Structural Atlas of the FSL package). We used the same images and object categories as in our experimental tasks (Experiment 1: cow, skate, dresser and flower; Experiment 2: cow, dresser, skate, butterfly, motorbike and tree). This localized object-selective region of interest corresponded to the posterior fusiform part of lateral occipital cortex (pFs), also referred to as posterior fusiform gyrus (pFG). Stimuli were presented in 24 blocks of images from the same category with each image shown for 250 ms. In Experiment 1, stimuli consisted of 48 intact objects (12 of each object category) and 48 scrambled objects (12 of each object category) that were presented in separate blocks for each object category (24 blocks in total, six per category) with fixation block intermixed (seven in total). In Experiment 2, because we also included the categories from the duplicate search task, we had 72 intact objects (12 per category) and 72 scrambled objects also presented in separate blocks for each object category (24 blocks in total, four per category).

In both Experiments, the mapper run lasted ~7 min. Participants were asked to push a button when two consecutive images were identical (same exemplar) to ensure attention. The same fMRI preprocessing steps as described for the experimental task were performed for this mapper. In Experiment 1, for two participants the data recorded from this mapper was not usable, therefore, we used the anatomical mask only for these participants. In Experiment 2, due to a programming error, for the first three participants the mapper only contained three categories (cow, skate and tree).

## fMRI acquisition

Scanning was done on a 3T Philips Achieva TX MRI scanner with a 32-elements head coil. In the middle of the testing session (after four runs) a high-resolution 3DT1-weighted anatomical image (TR, 8.35 ms; TE, 3.83 ms; FOV, 240 × 220 × 188, 1 mm3 voxel size) was recorded for every participant (duration 6 min).

During the experimental task an object-selective cortex functional localizer, blood oxygenation level dependent (BOLD)-MRI was recorded using Echo Planar Imaging (EPI) (TR 2000 ms, TE 27.62 ms, FA 76.1, 36 slices with ascending acquisition, voxel size 3 mm3, slice gap 0.3 mm, FOV 240 × 118.5×240).

## fMRI data analysis

### fMRI preprocessing

MRI data were registered to the subject-specific T1 scan using boundary-based registration (*Greve and Fischl, 2009*). The subject-specific T1 scan was registered to the MNI brain using FMRIB's Nonlinear Image Registration Tool (FNIRT). For the functional imaging data, we used FEAT version 5, part of FSL (Oxford Centre for Functional MRI of the Brain (FMRIB) Software Library; www. fmrib.ox.ac.uk/fsl; [*Smith et al., 2004*]). Preprocessing steps consisted of motion correction, brain extraction, slice-time correction, alignment, and high-pass filtering (cutoff 100 s). For each subject and each trial a general linear model (GLM) was fitted to the data, whereby every TR (2 s each) was taken as a regression variable. We derived the t-value of each voxel for each of the 15 (Experiment 1) or 16 (Experiment 2) TRs that were part of each trial in Experiment 1 and Experiment 2 respectively. We used FMRIB's Improved Linear Model (FILM) (*Woolrich et al., 2001*) for the time-series statistical analysis. The data were further analyzed in MATLAB (The MathWorks, Natick, MA, USA). For every participant, every run, every experimental condition (Experiment 1: Current, Prospective; Experiment 2: Current, Prospective and Irrelevant), category exemplar (Cow, Dresser and Skate, 4 exemplars of each) and for each TR, we created a vector containing the t-value per voxel in our regions of interest (see below). T-values for each predictor were computed by dividing the beta-

weight by the standard error. That vector comprised the spatial pattern of activity evoked at that time point (TR) for that experimental condition in our region of interest.

## Within-relevance and Cross-relevance object category decoding

Next, we used these multi-voxel patterns to answer the question whether Relevance (Experiment 1: current or prospective; Experiment 2: current, prospective, irrelevant) affected the neural category representations. To determine this we used the Princeton Multi-Voxel Pattern Analysis toolbox (available at https://github.com/princetonuniversity/princeton-mvpa-toolbox, see *Detre et al., 2006*). To examine whether current, prospective and irrelevant items evoked a distinct pattern of activity in pFs, for each condition and TR, a single class logistic regression classifier was trained to distinguish each object category (cow, dresser and skate). Logistic regression computes a weighted combination of voxel activity values, and it adjusts the (per-voxel) regression weights to minimize the discrepancy between the predicted output value and the correct output value. The maximum number of iterations used by the iteratively-reweighted least squares (IRLS) algorithm was set to 5000.

In Experiment 1, classifier performance was evaluated with a standard leave one run out cross validation procedure. This involved training a single class logistic regression classifier to learn a mapping between the neural patterns and the corresponding category labels for all but one run, and then using the trained classifier to predict the category of stimuli from the test patterns in the remaining run. For each iteration, we trained the classifier on seven runs and tested on the remaining run for each ROI. Overall classification accuracy was the average accuracy of the nine iterations.

In Experiment 2, relevance condition was fully balanced within each run; however, we could not fully balance the relevance condition by object category combinations within each run (see Task and stimuli of Experiment 2). Therefore, we used a modified leave one run out cross-validation procedure. Per run we had four trials per relevance condition; therefore, each training set consisted of 8 runs with 32 trials per relevance condition which is not a multiple of 3 (i.e., the amount object categories of interest). Thus, for each relevance condition (Current, Prospective Irrelevant) when selecting all but one run for the training set, one of the categories contained 10 exemplars while the other two categories had 11. Likewise, the testing set contained all three categories (i.e., cow, dresser, skate), but two of the four exemplars belonged to the category that was less frequent in the training set. To correct for this slight unbalance and ensure that the classifiers were not biased against the least frequent category, we picked one exemplar (for each of the two categories that had 11) and excluded them from the training set, leaving 30 exemplars per training set per relevance condition (10 per category). We repeated this entire process and left a different exemplar out of the training set, until all exemplars were left out exactly once and all exemplars were included an equal number of times across all train-test procedures. Therefore, for each run out, we trained and tested 11 classifiers (99 in total per TR). Overall classification accuracy was the average accuracy of these 99 iterations. Moreover, we used a balanced accuracy calculation as described in *Fahrenfort et al. (2018)*, where accuracy is calculated separated per class and then averaged across classes.

In both Experiments we ran two types of decoding. We investigated category decoding (Cow, Dresser and Skate) both *within* Current, Prospective and Irrelevant conditions (within-relevance decoding) and *between* relevance conditions (cross-relevance decoding) for each time point (TR) in the trial separately. For the within relevance classification, we trained and tested the classifier on the same condition (Current, Prospective or Irrelevant). For the cross-relevance classification in Experiment 1, we trained when the category was a Current item and tested when the category was a Prospective item ('PC') and vice versa ('CP'). In Experiment 2, we applied this same cross-relevance decoding scheme (Current-Prospective) and added two more: Current-Irrelevant and Prospective-Irrelevant. This resulted in six different testing and training combinations. To reduce the amount of comparisons needed, we averaged the classification performance of those combinations where the same conditions were used either for testing or training. All the significance testing was performed on the averaged data of Current-Prospective, Current-Irrelevant and Prospective-Irrelevant. We obtained a classification score (percentage correct) per participant for every relevance condition and time point (TR). Note that here chance decoding was 33.33% since we had three object categories (Cow, Dresser and Skate). All statistical comparisons are based on two-tailed tests, except for the comparison against chance in the within-relevance coding scheme as there decoding cannot go below chance (cf. *Christophel et al., 2018*). All statistical analyses were performed using SPSS 17.0 (IBM, Armonk, USA).

## Cross-temporal generalization of object category decoding

The main analyses of Experiment 1 were based on decoding performance where training and testing occurred separately for each TR. To examine whether the neural representations of the different object categories for current and prospective states were also related across time, we tested for cross-temporal generalization of decoding accuracy (see *King and Dehaene, 2014*), by training the classifier on each of the TRs and then testing it on all other TRs in the trial. This was then repeated for all TRs, creating a two-dimensional matrix of cross-temporal object category decoding (with no additional smoothing). Time windows of significant decoding were identified using 2-dimensional cluster-based permutation testing (i.e., across both time axes) with cluster correction ($p = 0.05$ and 10.000 iterations) to statistically compare the object category decoding with chance (33.33%) (*Maris and Oostenveld, 2007*) using and MATLAB (The MathWorks, Natick, MA, USA). As a result, we were able to assess the temporal stability of object category decoding and to test whether encoding and maintenance of the object (Delay) was similar to searching for an object (Search 1 and Search 2).

## Representational dissimilarity analysis

For each TR we created a representational dissimilarity matrix (RDM) (*Kriegeskorte et al., 2008*; *Kriegeskorte and Kievit, 2013*). Each cell of the matrix represents a 1-rho (Spearman correlation) of the activity patterns of two individual exemplars. In Experiment 1, the RDMs consisted of $24 \times 24$, four unique exemplars per category (Cow, Dresser and Skate) and two different relevance levels (Current and Prospective). In Experiment 2, we created three different sets of RDMs depending on the conditions correlated (Current-Prospective, Current-Irrelevant, Prospective-Irrelevant). The RDMs of each run were averaged to obtain one RDM per TR. We further averaged across the three TRs for each interval of interest in the trial (Delay, Search 1 and Search 2). For visualization purposes we transformed the RDM by replacing each element by its rank in the distribution of all its elements (scaled between 0 to 1). In addition, we used multidimensional scaling (MDS) plots wherein the distance between points reflects the dissimilarity in their neural patterns of response. To compute the interaction between Relevance and Category over the course of the trial we calculated the dissimilarity for the between Relevance (Experiment 1: Current-Prospective, Experiment 2: Current -Prospective; Current-Irrelevant; Prospective-Irrelevant) and Category (same [Cow/Cow, Dresser/Dresser and Skate/Skate] vs different [Cow/Dresser, Dresser/Skate, Skate/Dresser]) by averaging the cells within each class. We calculated this for every TR separately, and then averaged those across the three TRs in the predetermined intervals (Delay, Search 1 and Search 2).

## Acknowledgements

We thank Assaf Harel for providing the pictures used as stimuli. This research was supported by the European Research Council (ERC) under grant agreement no. ERC-CoG-2013-615423 awarded to CNLO.

## Additional information

### Funding

| Funder | Grant reference number | Author |
| --- | --- | --- |
| European Research Council | 615423 | Christian N L Olivers |

The funders had no role in study design, data collection and interpretation, or the decision to submit the work for publication.

### Author contributions

Anouk Mariette van Loon, Conceptualization, Data curation, Software, Formal analysis, Investigation, Visualization, Methodology, Writing—original draft, Project administration, Writing—review and editing; Katya Olmos-Solis, Conceptualization, Data curation, Formal analysis, Investigation, Visualization, Methodology, Project administration, Writing—review and editing; Johannes Jacobus

Fahrenfort, Conceptualization, Supervision, Methodology, Writing—review and editing; Christian NL Olivers, Conceptualization, Supervision, Funding acquisition, Investigation, Methodology, Writing—original draft, Writing—review and editing

### Author ORCIDs
Anouk Mariette van Loon (iD) http://orcid.org/0000-0002-9015-7647
Katya Olmos-Solis (iD) http://orcid.org/0000-0002-3191-2286
Johannes Jacobus Fahrenfort (iD) http://orcid.org/0000-0002-9025-3436
Christian NL Olivers (iD) http://orcid.org/0000-0001-7470-5378

### Ethics
Human subjects: Twenty-four healthy volunteers participated in Experiment 1 and twenty-five healthy volunteers participated in Experiment 2. The experiment was approved by the Ethical Committee of the Faculty of Social and Behavioral Sciences, University of Amsterdam and conformed to the Declaration of Helsinki. All subjects provided written informed consent and consent to publish.

### Decision letter and Author response
Decision letter https://doi.org/10.7554/eLife.38677.025
Author response https://doi.org/10.7554/eLife.38677.026

## Additional files

### Supplementary files
• Transparent reporting form
DOI: https://doi.org/10.7554/eLife.38677.021

### Data availability
All data generated or analyzed during this study are included in the manuscript and supporting files. Source data files including the code have been provided for Figures 2,3,4 and 5. fMRI data is made available via the open science framework: "Current and Future Goals Are Represented in Opposite Patterns in Object-Selective Cortex." Open Science Framework. May 31. osf.io/hcp47.

The following dataset was generated:

| Author(s) | Year | Dataset title | Dataset URL | Database and Identifier |
| --- | --- | --- | --- | --- |
| van Loon A, Olmos Solis K | 2018 | Current and Future Goals Are Represented in Opposite Patterns in Object-Selective Cortex | https://osf.io/hcp47/ | Open Science Framework, hcp47 |

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
