## [Decision Letter]

Thank you for submitting your article "Current and future goals are represented in opposite patterns in object-selective cortex" for consideration by *eLife*. Your article has been reviewed by three peer reviewers, including Floris P de Lange as the Reviewing Editor and Reviewer #1, and the evaluation has been overseen by Sabine Kastner as the Senior Editor. The following individual involved in review of your submission has agreed to reveal their identity: Brad Postle (Reviewer #2).

The reviewers have discussed the reviews with one another and the Reviewing Editor has drafted this decision to help you prepare a revised submission.

Summary:

van Loon et al. examine working memory representations of information that is either currently relevant (current) or will be relevant later (prospective). Using fMRI, they find that both types of working memory can be decoded from activity patterns in mid-level visual activity patterns. Strikingly, activity patterns were opposite for these two types of memory, with information that will be needed to accomplish a future goal, but not the most proximal one, represented in a format that is anticorrelated with format in which it will be represented when it is of highest relevance.

All three reviewers were enthusiastic about the aims of the study. There was however considerable debate about whether the results could indeed be interpreted as 'coding prospectively relevant information', or whether they could be equally well represent BOLD undershoot (reviewer 1) or attentional suppression (adaptation) of the unattended working memory (WM) item and/or class (reviewer 3).

Essential revisions:

It will be essential to satisfactorily rule out the alternative interpretations of the data put forward by reviewer 1 (point 1 and 2) and reviewer 3.

Reviewer #1:

van Loon et al. examine working memory representations of information that is either currently relevant (current) or will be relevant later (prospective). Using fMRI, they find that both types of working memory can be decoded from activity patterns in mid-level visual activity patterns. Strikingly, activity patterns were opposite for these two types of memory.

This is an interesting study, on the currently 'hot topic' of where and how visual information is maintained in working memory in the human brain. The paper is well written and coherent, and the analyses are expertly done and clear. I am however worried about some alternative explanations for the results that may ultimately render them rather hard to interpret.

1) Can the results be explained by BOLD undershoot?

During the prospective trials, the participants see an exemplar of the category of interest (e.g., a cow) and the flower, and then they see a cue that tells them to first focus on the flower. The null hypothesis (i.e., no sensory activity pattern for items that only become relevant later) would dictate that activity in the 'cow area' would simply decline, and become active again only during search display 2. Under this null hypothesis, cow voxels would have negative values during search display 1, which would explain why cross-classification results in below-chance performance.

The authors swiftly dismiss this possibility, because, as Figure 2—figure supplement 1 shows, overall BOLD signal in the prospective condition actually goes up. However, what is plotted here is mean BOLD signal of the entire pFS area. Naturally, the flower search display will generate a large BOLD signal (in e.g. flower voxels). I think it is quite plausible that the results are explained by a BOLD undershoot of the relevant voxels, which is masked by a BOLD activation by a concomitant activation of other visually response voxels.

2) Can the results be explained by systematic differences in search display 1 between current and prospective trials?

Another alternative explanation (of which I'm less sure it holds water, but I thought it may be good to mention it anyway) of the anti-decoding during search 1, is that prospective trials are trials in which the search 1 display is always the 'flower search display'. Current trials are trials in which something else than flowers is presented. Is it possible that the below chance decoding means that flowers don't resemble the other categories? In other words, is it possible that below chance decoding has nothing to do with the memorized target, but with the presented search display? (which is systematically different between conditions?)

3) Design choice

The authors designed their task such that one target was 'of interest' (could be of three categories) whereas the other was always exactly the same item (the flower). I am wondering whether this is a good design choice. One of the potential problems of this design choice is already mentioned in point 2. But it seems to create other forms of imbalance. For example, a 'prospective trial' may be more difficult than a 'current' trial. In prospective trials, there is a longer WM load, in current trials there is a reduced WM load (because the flower doesn't really require holding in WM, as behavioral results also show).

I don't really understand what advantage the current design has, over a design in which e.g. two exemplars would be shown from the three categories employed by the authors. This would be a more balanced design, and it would allow the authors to truly compare current vs. prospectively relevant items, in a design in which these trials are of equal difficulty.

Reviewer #2:

"Current and future goals are represented in opposite patterns in object-selective cortex," van Loon, Fahrenfort, Olivers. This manuscript introduces what may well come to be accepted as an important principle in the representational dynamics of information held in versus out of the focus of attention in visual working memory, with information that will be needed to accomplish a future goal, but not the most proximal one, represented in a format that is anticorrelated with format in which it will be represented when it is of highest relevance. The approach taken by the authors represents an important advance, in that it addresses *how* information is maintained in different states of priority, rather than "just" *where*. (Questions of 'where' can be interesting, but only if they give insight into/constrain models of mechanism, and this field has matured to the point where questions of 'where in the brain' no longer offer much of an advance.) Overall, the manuscript is very clearly written and argued, and my comments will mostly highlight points where clarity could be improved.

Figure 2A: It appears that, by TR4 following the cue, the prospective category is at baseline – if so, this could be consistent with a stimulus encoding response that then returns to baseline (as opposed to the prospective category being "actively" (my term, but the authors' implication) sustained across the initial delay).

Figure 3B: This is a nice visualization. It could provide more information if the individual exemplars were identified (as, e.g., 1, 2, 3, 4 rather than four undifferentiated green dots). What's not clear to me, however, is whether any quantitative and/or inferential information is conveyed by the lines that cross in the middle of each MDS plot for Search 1 and Search 2.

"Next we wanted to investigate what turns a currently relevant into a prospectively relevant representation." This misrepresents what was accomplished in the analyses that follow, which were more along the lines of 'tracking what transformation a representation undergoes when transitioning from currently to prospectively relevant." Indeed, it is noted in the Discussion that "A crucial question that our data does not answer is what the exact mechanism is behind this transformation."

For the analyses illustrated in Figure 4, more information is needed. First, what condition labels are used for the three sets of t-values in each plot? E.g., for "cow," are the betas from the cow regressor from the GLM used for all three plots (Current Same, Prospective Same, Prospective different)? If this is true, it's not surprising that there's very limited variance across voxels in the latter two conditions. Would the regression be more interesting if it was "fair" to the Different category by using its regressors? Also, how does this analysis reconcile with the idea of an opposite representation? E.g., why isn't the slope of Prospective Same of the same value as Current Same, but opposite in sign? A separate question is how interpretable the classifier weights are. With logistic regression, the importance of any given voxel can vary markedly depending on the level of sparsity that is imposed (i.e., depending on the lambda term in the regression model). Haynes's group has described a way to get unequivocal importance values from SVM, but I'm not familiar with a comparable metric for logistic regression. If such a diagnostic can be extracted from logistic regression it would be of considerable interest, and so should be detailed.

"We found that object category information was stronger for pFs, visual cortex and IPS than for RMF, whereas the impact of relevance (Current vs. Prospective) was apparent in all ROIs throughout the trial." This is an important point, and addresses a question of considerable current theoretical interest (and controversy), and so more information should be included – enough to support critical evaluation of this statement.

Reviewer #3:

van Loon and colleagues present the results of an elegant fMRI study designed to test for representational differences/similarities in coding scheme for items in working memory with different priority status (immediately relevant vs. prospectively relevant). The study is well-conceived, well conducted and well-motivated to tackle an important problem. The analyses are clear and appropriate, however the results are somewhat surprising. The authors report a literal reversal in pattern during search in the irrelevant display. Even more surprising, this occurred in both the first and second search position (i.e., even after the non-relevant item could be forgotten). The authors present these results very clearly, and provide at least two interpretations: top-down control (attention suppression?) or neural adaptation, favouring the latter.

A major question left unresolved is whether this adaptation (or top-down suppression) is item specific or general to the whole class of stimuli (category specific). Obviously this is a trickier analysis, but theoretically tractable and worth reporting whether or not it is possible. The results would be very informative, especially interpreting the effects within a working memory framework.

In general, the authors do a good job handling what feels like unexpected results; however, I think there is room to be a bit more upfront with the initial hypotheses. They are relatively clear that they did not expect a pattern reversal, and especially not during the second search. For clarity, I think it would be worth explicitly stating the initial hypotheses regarding the first delay period (presumably there is a case for qualitative different formats in preparation for search).

---

## [Author Response]

Reviewer #1:[…] This is an interesting study, on the currently 'hot topic' of where and how visual information is maintained in working memory in the human brain. The paper is well written and coherent, and the analyses are expertly done and clear. I am however worried about some alternative explanations for the results that may ultimately render them rather hard to interpret.1) Can the results be explained by BOLD undershoot?During the prospective trials, the participants see an exemplar of the category of interest (e.g., a cow) and the flower, and then they see a cue that tells them to first focus on the flower. The null hypothesis (i.e., no sensory activity pattern for items that only become relevant later) would dictate that activity in the 'cow area' would simply decline, and become active again only during search display 2. Under this null hypothesis, cow voxels would have negative values during search display 1, which would explain why cross-classification results in below-chance performance.The authors swiftly dismiss this possibility, because, as Figure 2—figure supplement 1 shows, overall BOLD signal in the prospective condition actually goes up. However, what is plotted here is mean BOLD signal of the entire pFS area. Naturally, the flower search display will generate a large BOLD signal (in e.g. flower voxels). I think it is quite plausible that the results are explained by a BOLD undershoot of the relevant voxels, which is masked by a BOLD activation by a concomitant activation of other visually response voxels.

We thank the reviewer for his supportive words. We agree that the original experiment did not allow us to fully exclude the possibility that below-chance decoding may have been caused by a voxel-specific BOLD undershoot following the initial sensory stimulation of the to-be-remembered target. To rule out this interpretation, we performed a second experiment that specifically addresses this and related concerns. As in our initial experiment, observers committed an object to memory and were cued with regard to the relevance of that object prior to performing the first search task. Critically though, we included a third condition that was identical to the Prospective condition, but in which the item was no longer relevant in the remainder of the trial. We explain the logic behind adding this Irrelevant condition in the introduction of Experiment 2:

“To address this, Experiment 2 sought to replicate and extend the main findings with a number of design changes. […] For the same reason, neither can any differences between irrelevant and prospective representations be attributed to passive, sensory-related adaptation or BOLD undershoot.”

Importantly, Experiment 2 showed below-chance decoding during the Search 1 interval for the Current-Prospective decoding scheme, and more strongly so than for the Current-Irrelevant scheme, for which classification performance did not reliably differ from chance (see Figures 4C and 4D).

In addition, the representational dissimilarity analyses of Experiment 2 show that while Prospective representations differ more from current representations when of the *same* category compared to when of a different category, such differential dissimilarity did not occur for irrelevant representations (see Figure 5C and 5F).

One might argue that because the memory item was not a target in the irrelevant condition, the initial BOLD response may have been weaker in the Irrelevant than in the Prospective condition, leading to a similarly weaker undershoot. However, in Experiment 2 there were no differences in the magnitude of the mean BOLD response during the delay between the Irrelevant and Prospective conditions (from TR1 to TR4 in the delay all ps > 0.075, see also Figure 4—figure supplement 1 and here) and therefore, it is reasonable to assume that any sensory-related undershoot would also be of the same magnitude.

In a similar vein, peak classification accuracy in the within relevance decoding scheme did also not differ across conditions during the delay; if anything, peak classification accuracy was somewhat stronger in the Irrelevant than in the Prospective condition in TR1 (*t*(24) =2.56, *p* = 0.017, *d* = 0.51, see Figure 4A). So if initial sensory encoding strength caused a subsequent representation-specific undershoot, this should have been at least equally the case for irrelevant items.

Taken together, we believe that the inverse representational pattern observed in both experiments cannot primarily be explained by BOLD undershoot, and that additional mechanisms are involved when maintaining an item for later use (as compared to items that have become completely irrelevant). We now write in the Discussion:

“An alternative possibility is that local, and arguably more passive mechanisms cause the change in responsiveness. […] Thus, the representational differences we find are determined by the observer’s goal state, and not automatically induced by the sensory representation of the stimulus. “

2) Can the results be explained by systematic differences in search display 1 between current and prospective trials?Another alternative explanation (of which I'm less sure it holds water, but I thought it may be good to mention it anyway) of the anti-decoding during search 1, is that prospective trials are trials in which the search 1 display is always the 'flower search display'. Current trials are trials in which something else than flowers is presented. Is it possible that the below chance decoding means that flowers don't resemble the other categories? In other words, is it possible that below chance decoding has nothing to do with the memorized target, but with the presented search display? (which is systematically different between conditions?)

We fully understand this comment. First, let us remark that although the design of the original experiment may appear suboptimal in some ways, we explicitly chose it to be optimal in other ways. Specifically, we chose the categories of interest (cow, dresser, skate) to be the same for each subject and condition on the basis of earlier studies showing that these categories could be well dissociated (Harel, Kravitz, and Baker, 2014), and were moreover different from the flower category. We did this with the purpose to maximize the chance of decoding the item of interest even when prospective. Moreover, keeping the categories the same for every subject facilitated the RDM analyses. Second, note that our dependent measure was classification performance across the three categories of interest (cow, dresser, skate), while the flower remained constant regardless of which of these three categories was in memory. Classification performance, whether below chance or above chance, could thus not have been caused by the flower per se. What could have happened though is that the flower differentially *interacted* with any one of the categories of interest, leading to a particularly strong bias against one or two of the categories, which then shows in the average.

We therefore believed it would be prudent to also tackle this issue head on in Experiment 2, by replacing the flower search with a search for a range of new objects, i.e. trees, butterflies, and motor bikes. Each of these categories was paired with each category of the variable template (i.e. cow, dresser or skate). Moreover, the category sets for each type of tasks had one animate and two inanimate categories, making them more comparable. We also changed the other search task – which was not of interest – from a constant template task to one in which no template was needed. We briefly explain the main design changes in the introduction of Experiment 2:

“Furthermore, we simplified the design by keeping the variable template search (here referred to as simply “template search”), but replacing the constant template task of Experiment 1 [i.e. the flower search] with what we call a “duplicate search” task, in which participants indicated whether or not one of the objects in the search display appeared twice (see Figure 1B and Materials and methods for details, as well as de Vries et al., 2018). […] Finally, just like the stimuli for the template search were drawn from three different categories (cows, skates, and dressers), we varied the stimuli in the duplicate search such that they were also drawn from three different categories (specifically butterflies, motorcycles, and trees), in order to assess whether the (below-chance) decoding of prospective representations during search generalizes across a range of different categories.”

Note that we still decided to keep the primary objects of interest for our decoding and RDM analyses the same (cows, dressers, skates), as a) they worked in Experiment 1, and b) the category-specific RDM analyses require the categories to be identical across subjects.

Given that we found the same pattern of results as in Experiment 1, we believe our results were not specific to the flower, or in general to the use of a single search category when maintaining a prospective item. Furthermore, the difference between the Prospective and the Irrelevant condition (see earlier point) also argues against the effects being specific to the categories used.

3) Design choiceThe authors designed their task such that one target was 'of interest' (could be of three categories) whereas the other was always exactly the same item (the flower). I am wondering whether this is a good design choice. One of the potential problems of this design choice is already mentioned in point 2. But it seems to create other forms of imbalance. For example, a 'prospective trial' may be more difficult than a 'current' trial. In prospective trials, there is a longer WM load, in current trials there is a reduced WM load (because the flower doesn't really require holding in WM, as behavioral results also show).I don't really understand what advantage the current design has, over a design in which e.g. two exemplars would be shown from the three categories employed by the authors. This would be a more balanced design, and it would allow the authors to truly compare current vs. prospectively relevant items, in a design in which these trials are of equal difficulty.

We agree with the reviewer that we did not clearly explain in the manuscript the reasoning behind the design choices in Experiment 1, in particular, the reason why we used systematically different categories for the current and the prospective items within each trial. See also our response to the previous point. We now make our reasoning explicit in the Materials and methods section of Experiment 1:

“Notice that in Experiment 1, the current and prospective templates were always drawn from separate category sets within a trial. […] If both templates were to be drawn from the same category set within the trial, it would be impossible to know whether category classification accuracy actually reflects the quality of the representation of the current template, the prospective template or a combination of the two.”

In Experiment 1, we used a constant template search task (i.e., the flower task), as a way to limit memory load and to maximize the chances of decoding the category of interest when prospective (see Results section of Experiment 1). However, we also agree with the reviewer that this in itself created a difference in difficulty between the two types of search tasks. In Experiment 2, we replaced the flower search with a duplicate search task that we have successfully used before (De Vries et al., 2018), with one of the reasons being an attempt to obtain comparable levels of difficulty across the different type of task. Behavioral results indeed showed that there were no significant differences in participants’ accuracy between the two types of task, and that RT differences now went in the opposite direction compared to Experiment 1. Yet the fMRI results remained the same, and thus cannot be explained by differences in difficulty across tasks.

Finally, because the duplicate search itself requires no template, in Experiment 2 memory load was always one. It is still true though that, compared to the current condition, in the prospective condition participants held the item in memory for longer throughout the trial. Nevertheless, behavioral results in Experiment 2 also showed that participants were equally accurate at finding the template object regardless of the search order, albeit they were slower when the template search was performed second (i.e., Prospective condition). This suggests that the quality of the memory representation did not decay when it was prospective (and it had to be maintained for longer), but that retrieving it for Search 2 required additional time. So, any differences between the current and the prospective representations cannot be explained by participants being worse in remembering the prospective item.

Reviewer #2:"Current and future goals are represented in opposite patterns in object-selective cortex," van Loon, Fahrenfort, Olivers. This manuscript introduces what may well come to be accepted as an important principle in the representational dynamics of information held in versus out of the focus of attention in visual working memory, with information that will be needed to accomplish a future goal, but not the most proximal one, represented in a format that is anticorrelated with format in which it will be represented when it is of highest relevance. The approach taken by the authors represents an important advance, in that it addresses how information is maintained in different states of priority, rather than "just" where. (Questions of 'where' can be interesting, but only if they give insight into/constrain models of mechanism, and this field has matured to the point where questions of 'where in the brain' no longer offer much of an advance.) Overall, the manuscript is very clearly written and argued, and my comments will mostly highlight points where clarity could be improved.

We thank the reviewer for his appreciation of our work and are pleased to know that he finds our approach to be an important advance and the manuscript well written and argued. We hope that the changes to the manuscript have sufficiently improved its clarity.

Figure 2A: It appears that, by TR4 following the cue, the prospective category is at baseline – if so, this could be consistent with a stimulus encoding response that then returns to baseline (as opposed to the prospective category being "actively" (my term, but the authors' implication) sustained across the initial delay).

We agree with the reviewer that in Experiment 1 the classification performance for the prospective condition returned to baseline at TR4 and indeed we found significant differences between the Current and Prospective conditions at this TR. We now make this return to baseline explicit in the Results section of Experiment 1:

“[…] during the Delay prior to search the within-relevance decoding resulted in significant above chance object category decoding both when the variable template was currently relevant (*t*(1,23) = 8.18, *p* < 0.001, *d* = 1.67) and when prospectively relevant (*t*(1,23) = 5.67, *p* < 0.001, *d* = 1.16). […] Notably, decoding performance was higher when the item was currently relevant than when it was prospectively relevant (Current vs. Prospective: *t*(1,23) = 3.22, *p* = 0.004, *d* = 0.66), consistent with its importance for the upcoming search task.”

And we explicitly mention the return to baseline levels in the Discussion:

“The reduced activity account is partially supported by our data. We found that during the delay period prior to the first search task, before the evidence for the prospective item diminished to baseline levels, current and prospective representations were highly similar, as evidenced by a strong correlation and successful cross-relevance classification of the current, prospective and irrelevant representations.”

Figure 3B: This is a nice visualization. It could provide more information if the individual exemplars were identified (as, e.g., 1, 2, 3, 4 rather than four undifferentiated green dots). What's not clear to me, however, is whether any quantitative and/or inferential information is conveyed by the lines that cross in the middle of each MDS plot for Search 1 and Search 2.

We thank the reviewer for pointing out this lack of clarity in our plots. In the new version of this figure, instead of using the same symbols for all the exemplars of the same category, we used different symbols for each exemplar within a category. The categories are still indicated by color, while the relevance conditions are still indicated by the saturation. For instance, in the new Figure 3B, cows are purple, dressers are green and skates are orange. The light and dark purple triangles are then the same exemplar, when respectively current and prospective. The same applies for the other categories.

The lines that crossed in the middle of each MDS plot in the original figure were used as way to facilitate the readability of the plots, but they did not have any quantitative information; therefore we decided to remove them from the graph. The quantitative information can be found in Figure 3C, where we depict the mean dissimilarity values for current and prospective targets when drown from the same vs. different object categories.

"Next we wanted to investigate what turns a currently relevant into a prospectively relevant representation." This misrepresents what was accomplished in the analyses that follow, which were more along the lines of 'tracking what transformation a representation undergoes when transitioning from currently to prospectively relevant." Indeed, it is noted in the Discussion that "A crucial question that our data does not answer is what the exact mechanism is behind this transformation."

We agree with the reviewer that ‘transformation’ is a better-suited term. We have changed this line accordingly throughout the manuscript. For example:

“We find a pronounced anti-correlated representation for prospective targets compared to the same targets when current. Experiment 2 furthermore shows that it is not the temporary irrelevance, but the prospective relevance that causes this transformation, because little to no anti-correlation was found for items that were irrelevant altogether."

And:

“What causes current and prospective representations to anti-correlate? One possibility is that the brain separates current from prospective templates within the same neuronal ensembles by actively transforming the representational pattern of prospective templates to be opposite to that of current templates.”

For the analyses illustrated in Figure 4, more information is needed. First, what condition labels are used for the three sets of t-values in each plot? E.g., for "cow," are the betas from the cow regressor from the GLM used for all three plots (Current Same, Prospective Same, Prospective different)? If this is true, it's not surprising that there's very limited variance across voxels in the latter two conditions. Would the regression be more interesting if it was "fair" to the Different category by using its regressors? Also, how does this analysis reconcile with the idea of an opposite representation? E.g., why isn't the slope of Prospective Same of the same value as Current Same, but opposite in sign? A separate question is how interpretable the classifier weights are. With logistic regression, the importance of any given voxel can vary markedly depending on the level of sparsity that is imposed (i.e., depending on the lambda term in the regression model). Haynes's group has described a way to get unequivocal importance values from SVM, but I'm not familiar with a comparable metric for logistic regression. If such a diagnostic can be extracted from logistic regression it would be of considerable interest, and so should be detailed.

To address the reviewer’s request for clarification; reviewer 2 is correct to assume that the t-values for each predictor were indeed obtained by dividing the betas from the regressors in the GLM by the standard error. However, each line in the graph represented the t-values of a category when current (e.g. cow), the t-values for the condition in which the same category was prospective (here the cow, when in memory for the second task) and the average t-values of the conditions when one of the other two categories was prospective (i.e., here when either a dresser or a skate was in memory for the second task).

However, we decided to remove this analysis from the manuscript altogether since we agree with the reviewer that the interpretation of classifier weights for logistic regression is not straightforward (Haufe et al., 2014). Moreover, given that largely the same information was already conveyed by the RDMs, we believe that removing the analysis from the manuscript does not weaken the main conclusions of the paper.

"We found that object category information was stronger for pFs, visual cortex and IPS than for RMF, whereas the impact of relevance (Current vs. Prospective) was apparent in all ROIs throughout the trial." This is an important point, and addresses a question of considerable current theoretical interest (and controversy), and so more information should be included – enough to support critical evaluation of this statement.

We agree with the reviewer on the importance of this finding. However, the addition of Experiment 2 already made the manuscript very dense, especially if we also included the same analyses again there. Therefore, we decided to remove the analyses on other ROIs (visual, IPS and RMF) from the manuscript. Following suggestions of reviewer 3, we plan to first analyze these regions more thoroughly and then report on them in a separate paper.

Reviewer #3:[…]A major question left unresolved is whether this adaptation (or top-down suppression) is item specific or general to the whole class of stimuli (category specific). Obviously this is a trickier analysis, but theoretically tractable and worth reporting whether or not it is possible. The results would be very informative, especially interpreting the effects within a working memory framework.

We thank the reviewer for the supportive words, and for this interesting question. Unfortunately, to properly answer it, we would need to run our classifications schemes separately for each exemplar and, given the limited number of trials per exemplar and condition in our design, we do not have enough data to run such analysis. Also, in Experiment 1, while the categories were balanced across all cells of the design, the specific exemplars were not. In Experiment 2 we did balance the exemplars as well, but this resulted in only 3 trials per exemplar and condition combination. We now explicitly mention in the Discussion that this question is an unresolved issue that is worthy of future investigation:

“An issue left unresolved in the present study is whether the observed transformation of prospective representations extends to specific exemplars. […] Unfortunately, the limited number of trials per exemplar and condition precluded a useful analysis here, and future studies should directly test the item-specificity of the transformation of prospective representations.”

In general, the authors do a good job handling what feels like unexpected results; however, I think there is room to be a bit more upfront with the initial hypotheses. They are relatively clear that they did not expect a pattern reversal, and especially not during the second search. For clarity, I think it would be worth explicitly stating the initial hypotheses regarding the first delay period (presumably there is a case for qualitative different formats in preparation for search).

We agree with the reviewer that in the first version of the manuscript our hypothesis could have been more clearly stated. We want to emphasize that we explicitly planned the cross-relevance decoding scheme, and that we hypothesized the current and prospective items to be different (dissimilar) – but not in the way we observed. Now we explicitly write our initial hypothesis for all time intervals in Experiment 1:

“[…] while the within-relevance decoding scheme provides evidence for the presence of current and prospective representations, it does not reveal whether these representations are similar or different. […] Our general starting hypothesis was that while current and prospective representations would be similar during encoding, they would become increasingly dissimilar during the course of the trial, due to reduced activity or re-coding of the prospective item within the same network, while becoming similar again when the prospective memories are revived for the second task.”

When discussing the below-chance decoding results of Experiment 1, we then make clear what we did, and did not expect to find:

*“*This then raises the question as to whether the re-emerging prospective representation resembles its counterpart when currently relevant. […] The reliable deviation from chance further confirms that information on the prospective memory was present in object-selective cortex during the first search.”

Things change for Experiment 2 of course, as we now had explicit predictions derived from Experiment 1. Moreover, in the introduction of Experiment 2 we explicitly mentioned the ideas about possible cognitive mechanisms (or possible stimulus induced effects) underlying the transformation of the prospective items as observed in Experiment 1, which fed into our design changes:

*“*What causes current and prospective representations to anti-correlate? […] Specifically, presenting the to-be-memorized stimulus may result in neural adaptation (Henson and Rugg, 2003; Larsson and Smith, 2012; Vautin and Berkley, 1977) or in a BOLD-related undershoot (Huettel and McCarthy, 2000), each of which would predict a reduced voxel response to later stimulation.”